# ORCaS: Unsupervised Depth Completion via Occluded Region Completion as Supervision

**Hyoungseob Park[1], Runjian Chen[2], Patrick Rim[1], Dong Lao[3], Alex Wong[1]** *

[1]Yale University, [2]University of Hong Kong, [3]Louisiana State University

## Abstract

We propose a method for inferring an egocentric dense depth map from an RGB image and a sparse point cloud. The crux of our method lies in modeling the 3D scene implicitly within the latent space and learning an inductive bias in an unsupervised manner through principles of Structure-from-Motion. To force the learning of this inductive bias, we propose to optimize for an ill-posed objective during training: predicting latent features that are not observed in the input view, but exist in the 3D scene. This is facilitated by means of rigid warping of latent features from the input view to a nearby or adjacent (co-visible) view of the same 3D scene. "Empty" regions in the latent space that correspond to regions occluded from the input view are completed by a Contextual eXtrapolation (ConteXt) mechanism based on features visible in input view. The learned inductive bias of ConteXt can be transferred to modulate the features of the input view to improve fidelity. We term our method "Occluded Region Completion as Supervision" or *ORCaS*. We evaluate ORCaS on VOID1500 and NYUv2 benchmark datasets, where we improve over the best existing method by 8.91% across all metrics. ORCaS also improves generalization from VOID1500 to ScanNet and NYUv2 by 15.7% and robustness to low density inputs by 31.2%.

## 1 Introduction

Depth completion is the task of inferring an egocentric 2.5D dense depth map from a set of sparse points and an RGB image. The mechanism behind this process can be interpreted in two ways: (1) it propagates depth values from a set of sparse points to a denser lattice defined by pixels, while using the image as a condition to guide propagation; or (2) it uses the image to reconstruct a scale-ambiguous dense depth map, while using sparse depth values to calibrate the scale of the reconstruction. While they might appear to be merely two symmetric perspectives describing the same functional mapping, and the roles of the two input modalities seem superficially interchangeable, they differ fundamentally in the underlying principles. The former, (1), can be conceptualized as interpolation, leveraging natural image statistics (e.g., color, texture, edges) as regularization and therefore does not require induction. The latter, (2), on the contrary, relies on induction, as it attempts to impute a 3D scene from a single view, which is inherently ill-posed.

In general, (1) does not require learning, e.g., it is sufficient with handcrafted rules (Ku et al., 2018), but, if one wishes, can be easily learned by networks with limited capacity (Wong et al., 2020). However, it quickly faces saturation as one attempts to generalize the methodology to novel 3D scenes. Hence, it becomes inevitable to shift focus towards (2), whose ill-posedness necessitates learning an inductive bias – through which we subscribe to unsupervised learning, as ground truth required for supervised learning is expensive to acquire. The training signal comes from minimizing reconstruction error of the observed (input) image and sparse points by means of rigid warping from other (adjacent) views of the same 3D scene, e.g., Structure-from-Motion. Any region with sufficiently exciting textures that are co-visible between the input and adjacent views can be corresponded, while homogeneous regions and occluding boundaries are ambiguous and cannot be uniquely determined. Generic regularizers, such as local smoothness, are typically employed to learn the induction bias.

---
*{hyoungseob.park, patrick.rim, alex.wong}@yale.edu
{rjchen}@cs.hku.hk, {dong.lao}@lsu.edu

Yet, these regularizers are akin to those in (1); hence, what would be learned is the use of image for guided propagation. Instead, we consider a different supervision signal that cannot be modeled by generic regularizers: regions occluded from observed input view, which necessitates a stronger inductive bias beyond that of the 2D image, and of the 3D scene. We hypothesize that incorporating this as a learning objective will lead to higher fidelity predictions for egocentric depth completion.

One may question, given that depth completion only requires estimating depth for *visible surfaces*, how tasking the model to predict occluded regions (i.e., what is *not visible*) could aid in the reconstruction. Predicting occluded regions facilitates learning representations of the observations in 3D as opposed to typical 2D feature maps (Wong et al., 2021a; Wong & Soatto, 2021; Ma et al., 2019; Lopez-Rodriguez et al., 2020; Yan et al., 2023) of visible regions. This offers a few advantages: given the shape of an "object" in 3D, attributing (metric) scale requires only a single sparse point, allowing one to be less sensitive to the density of the sparse point cloud; additionally, predicting unseen portions of the 3D scene also facilitates learning higher levels of abstraction, e.g., "objects", which improves generalization.

To this end, we propose **O**ccluded **R**egion **C**ompletion **a**s **S**upervision (**ORCaS**) for unsupervised depth completion. ORCaS is a simple-yet-effective framework to enable learning from occluded regions in an input view. Like existing unsupervised depth completion methods, we encode the inputs as 2D features maps, but predict a probability distribution over depth planes for each feature vector and broadcast the features into a 3D volume through an orthogonal backprojection. During training, given an image and sparse depth map of an input view, its adjacent view, and a relative pose matrix between the two views, we perform a rigid warping to transfer the 3D features from the input view to the adjacent view. As the 3D features will only populate the co-visible regions between the two views, the "empty" regions could be empty or occupied by surfaces. Our method learns a set of parameters that populate the empty feature regions based on their location. When used to modulate the 3D features belonging to the input view (e.g., a single image and sparse depth map) at test time, ORCaS serves as an inductive bias and augments the volume based on its context. When the 3D features are mapped back to 2D, they can be seamlessly decoded to an egocentric dense depth map.

Training ORCaS is straight-forward; like existing unsupervised methods, we also assume access to image and sparse depth pairs of adjacent (forward and backward) views. However, rather than only reconstructing input image and sparse depth map from adjacent views, we also reconstruct the adjacent views by predicting their features from the input view using ORCaS. This naturally translates to supervision signals in both the latent feature and output spaces, and can be trained end-to-end in an alternating fashion, where we optimize the entire network in one alternation and only the parameter of ORCaS in another. While we utilize relative pose between input and adjacent views during training, we operate with the same input requirements as standard depth completion methods at inference: an RGB image and sparse depth map.

**Our contributions**: We propose (1) a novel supervision signal for unsupervised depth completion – to the best of our knowledge, we are the first to exploit regions occluded from the input view as means of learning an inductive bias for depth completion. This is made possible by (2) a simple-yet-effective architecture that enables transformation of features to adjacent views for learning parameters of ORCaS, which is used to modulate features of input view to improve fidelity. To do so, we introduce (3) ORCaS loss function to force the learning of the inductive bias in an alternating fashion. (4) Our method improves the state-of-the-art unsupervised depth completion on the VOID1500 and NYUv2 benchmarks by an average of 8.91%. ORCaS also demonstrates superior generalization, improving zero-shot transfer from VOID1500 to NYUv2 and ScanNet by an average of 15.7% and performance on low-density inputs on VOID150 by 31.2%.

## 2    RELATED WORK

**Supervised depth completion** approaches utilize the ground truths from range sensors (e.g., ToF, Light, Stereo cameras and LiDAR). (Huang et al., 2019; Uhrig et al., 2017) craft sparsity-invariant convolution layers to preserve sparse details. *Guided Depth Completion* supplies RGB image as a secondary input. (Jaritz et al., 2018) late-fuses dense RGB and depth. (Li et al., 2020) utilizes multi-scale processing through a cascade hourglass network. (Yang et al., 2019; Eldesokey et al., 2018; 2020; Ezhov et al., 2024; Qu et al., 2021; 2020) leverge the uncertainty of prediction. (Qiu et al., 2019; Xu et al., 2019; Zhang & Funkhouser, 2018) use surface normals to refine the depth predic-

tion. (Merrill et al., 2021; Sartipi et al., 2020; Zuo et al., 2021) capitalize on SLAM/VIO's camera data. (Krishna & Vandrotti, 2023) takes temporarily into consideration. Affinity-based frameworks are developed to refine depth map prediction. Spatial Propagation Networks (SPNs) (Liu et al., 2017; Cheng et al., 2018; 2020; Park et al., 2020; Lin et al., 2022) utilize learned affinity matrix to propagate the dense depth. (Chen et al., 2019b) presents a 2D-3D feature fusion. (Kam et al., 2022) is capable of presenting a richer scene topology by lifting 2D feature up to 3D representation. They process 3D volume features obtained by 3D point cloud and 2D RGB image embedding. (Yan et al., 2024) proposes a tri-perspective view decomposition strategy that explicitly encodes multi-view geometric cues to enhance geometry-aware depth completion. (Liang et al., 2025) distills the foundation monocular depth estimation model's prediction to train the depth completion model. (Yan et al., 2025) proposes a degradation-aware, selectively image-guided network that formulates depth completion as an enhancement problem to remain robust under degraded RGB observations. (Zuo et al., 2025) introduces Omni-DC, which integrates multi-resolution depth representations to achieve highly robust depth completion under varying sparsity and noise conditions. (Wang et al., 2025a) proposes PacGDC, a label-efficient and generalizable depth completion framework that leverages projection ambiguity modeling and consistency constraints to improve cross-domain performance. (Singh et al., 2023; Rim et al., 2026) extend depth completion to radar points.

**Unsupervised depth completion** (Ma et al., 2019) designs an early fusion, self-supervised training framework using Perspective-n-Point (PnP) (Lepetit et al., 2009) with Random Sample Consensus (RANSAC, (Fischler & Bolles, 1981)) and pose estimation to deduce photometric consistency loss. (Van Gansbeke et al., 2019) proposes the late fusion of global and local branch features to refine the depth prediction. (Shivakumar et al., 2019) leverages a depth prior learned using supervised training on an additional dataset. (Yang et al., 2019) learns a prior on shapes found in synthetic scenes, while (Lopez-Rodriguez et al., 2020) uses low-level features learned from synthetic data as guidance for the real domain. (Wong et al., 2020; 2021a) have proposed lightweight, VIO-compatible frameworks with dense input depth achieved by Scaffolding (Wong et al., 2020) and Spatial Pyramid Pooling (SPP, (He et al., 2015)) trained on synthetic scenes (Wong et al., 2021a). A line of studies have utilized *3D feature*. (Wong & Soatto, 2021) upgrades SPP to Sparse-2-Depth and imposes an inductive bias of backprojecting the feature representations onto RGB 3D space using approximated depths and input camera intrinsic matrix. (Yan et al., 2023) learns relative depth and predicts absolute scale separately. (Jeon et al., 2022) leverages line features rather than point features from visual SLAM. (Liu et al., 2022) distills knowledge from a blind ensemble of teachers by selecting the teachers that minimize reconstruction error. (Yu et al., 2023) uses self-attention for encoding and cross-attention for one-pass depth propagation. (Wu et al., 2024) proposes a framework to enable use of previously inviable photometric and geometric augmentations. Chung et al. (2025); Park et al. (2024) propose unsupervised test-time adaptation and (Chen et al., 2024; Rim et al., 2025) to continual learning. (Chancán et al., 2025) uses reprojection error for active learning.

The unsupervised methods have employed multi-view images to supervise depth and pose through reconstruction losses. Reconstruction relies on inverse warping, which projects the only "visible" points from the adjacent views onto the input view, discarding the occluded regions. In contrast, our method leverages the "invisible" regions-beyond what is available in the input view-by predicting the occluded region's feature in the adjacent views from the co-visible regions, thus improving the predictions for the input view. The adjacent views' features provide additional supervision, guiding the prediction of the adjacent view from the co-visible region's representation of the input view. This inductive process further improves the proposed method ORCaS's performance on input view.

**Multiple Plane Images (MPIs)**. The previous works utilizing MPI (Tucker & Snavely, 2020; Zhao et al., 2022; Abdelkareem et al., 2023) shares the similar vein of idea broadcasting and warping 2D features to discrete 3D planes. While MPIs have been primarily used in synthesizing images, our method learns an inductive bias with occlusion prediction as a regularizer for depth completion.

**Reconstructions of the occluded regions.** (Lao & Sundaramoorthi, 2018; Lao et al., 2021) show that occlusion in 2D image sequences can be reconstructed through a layered approach. In 3D, (Tulsiani et al., 2018) infer a layer-structured scene representation from a single image using view synthesis as supervision. (Dhamo et al., 2019) regress layered depth images from a single RGB input. (Kulkarni et al., 2022) introduces directed ray distance functions to reconstruct full 3D scenes from a single view, including occluded regions. (Wimbauer et al., 2023) predicts single-view 3D density fields supervised by multi-view photometric consistency. X-Ray (Hu et al., 2024) is a sequential ray-based 3D representation that encodes multi-layer surface frames to support diffusion-based 3D

generation of both visible and hidden object surfaces from images or text. (Li et al., 2024) uses spatial vision-language reasoning to enrich 3D point features with semantic context and language-guided attention. (Li et al., 2025) models multiple surface intersections per ray via layered point maps and a ray-stopping index. RaySt3R (Duisterhof et al., 2025) formulates shape completion as novel-view synthesis via a feed-forward transformer to predict depth, masks, and confidence along query rays.

## 3 METHOD FORMULATION

Given an RGB image $I : \Omega \subset \mathbb{R}^2 \to \mathbb{R}^3_+$, where $\Omega$ is the image domain, and its synchronized sparse point cloud $z : \Omega_z \subset \Omega \to \mathbb{R}_+$ projected onto the image plane, the depth completion aims to learn the function $\hat{d} = f(I, z)$ that reconstructs a dense depth map $\hat{d} : \Omega \subset \mathbb{R}^2 \to \mathbb{R}_+$ of the 3D scene.

Unsupervised depth completion leverages photometric reconstruction objectives and sparse depth consistency as supervision signals. Following recent approaches (Wong et al., 2020; Wong & Soatto, 2021), we assume (1) an input pair of RGB image and associated sparse depth map $(I_t, z_t)$ captured at input view $t$ and (2) an adjacent view $\tau$, where $\tau \in \{t - 1, t + 1\}$ provides sufficient parallax and co-visibility to view $t$. The reconstruction $\hat{I}_{t\leftarrow\tau}$ is obtained by reprojecting image $I_\tau$ into the image $I_t$'s view, using the predicted depth $\hat{d}_t := f(I_t, z_t)$ and the relative camera poses $g_{\tau\leftarrow t} := \rho(I_t, I_\tau)$ between adjacent views and the input view, where $\rho(\cdot)$ estimates the camera pose:

$$\hat{I}_{t\leftarrow\tau}(x) = I_\tau(\pi g_{\tau\leftarrow t} K^{-1} \bar{x} \hat{d}_t(x)), \tag{1}$$

where $g_{\tau\leftarrow t}$ denotes the relative camera pose matrix from time $t$ to time $\tau$, $\bar{x}$ the homogeneous coordinates of $x \in \Omega$, $K \in \mathbb{R}^{3\times3}$ the camera intrinsic calibration matrix, and $\pi$ the canonical perspective projection. Using this reconstructed image, a depth completion network $f_\theta$ minimizes:

$$\arg\min_\theta \sum_{\tau \in T} \sum_{x \in \Omega} \lambda_I \mathcal{P}\big(\hat{I}_{t\leftarrow\tau}(x), I_t(x)\big) + \sum_{x \in \Omega_z} \lambda_z \psi\big(\hat{d}_t(x), z_t(x)\big) + \lambda_r R(I_t, \hat{d}_t), \tag{2}$$

where $\mathcal{P}$ denotes the photometric reconstruction objective that minimizes the $L_1$ difference in pixel values and structural similarity (SSIM), $\psi$ the sparse depth reconstruction error, and $R$ the smoothness regularization objective that biases the depth map to be piece-wise smooth with discontinuities aligned with edges in the image, following (Ma et al., 2019; Wong et al., 2020; Wong & Soatto, 2021). $\lambda_I$, $\lambda_z$ and $\lambda_r$ are the weightings for their respective loss terms.

### 3.1 MOTIVATION

3D reconstruction is an ill-posed problem; hence, its solution hinges on the choice of regularizers or assumptions made about the 3D scene. While one can employ generic (hand-crafted) regularizers, such as local smoothness conditioned on intensity changes within the image Ma et al. (2019); Wong et al. (2020); Wong & Soatto (2021), regularities are imposed up to the appearance patterns present in the image: They may correspond to a discontinuity within the 3D scene or just the textures of an object. Hence, there is a need to force the learning of higher levels of abstractions, such as the shape of the objects. In order to learn this, we consider the under-constrained task of predicting or completing occluded regions from an observed view. Because occluded regions, by definition, are not visible, it necessitates an inductive process, where the underlying latent variable shared across projections of objects onto 2D images is the 3D object itself. We hypothesize that the inductive bias learned can be used to enrich the features of observations to aid completion of "missing" points – which is precisely the task of depth completion.

### 3.2 ORCAS ARCHITECTURE

To facilitate the learning of this inductive bias, we aim to predict the occluded regions in view $t$ that correspond to visible regions in view $\tau$, given an input RGB image and sparse depth map in view $t$. To achieve this, we pass them through an encoder to extract 2D feature maps. Given the depth imputed from these feature maps, we (i) backproject them into a 3D volume composed of depth planes. The 3D volume is then (ii) rigidly warped from the input view $t$ to an adjacent view $\tau$ using the relative pose $g_{\tau\leftarrow t}$. Note that the warped volume covers only the regions co-visible in both views

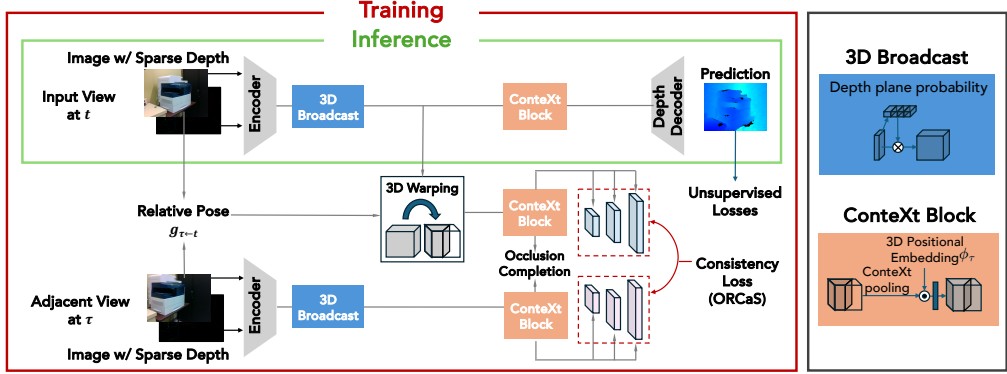

Figure 1: *Overview of Occluded Region Completion as Supervision (ORCaS)*. Inference of ORCaS for the input view only requires a single input view ($t$), and an identity camera pose matrix. Training ORCaS involves two different views (input view $t$, and target view $\tau$) and their relative camera pose $g_{\tau \leftarrow t}$. The input view 3D features are warped to align with the adjacent view. Empty regions due to occlusion are predicted by the ConteXt layer, and the inductive bias is learned by minimizing ORCaS loss, which leverages the extracted 3D feature from the adjacent view inputs as supervision.

$t$ and $\tau$, leaving regions occluded in $t$ "empty" in $\tau$. Naturally, (iii) the task of reconstructing the empty regions from the co-visible regions emerges as an auxiliary supervision signal. To this end, (iv) an RGB image and sparse depth map in an adjacent view $\tau$ can also be encoded to be directly used as supervision through our proposed ORCaS loss function during training. The overview of the ORCaS is illustrated in Fig. 1. We detail these steps below.

**(i) Broadcasting 2D features to 3D voxels**. To backproject 2D features into a 3D volume, we model the discrete probability distribution of depth at position $x$ over the $D$ depth planes as $\tilde{d}[x]$. Given $D$ uniformly distributed depth planes, each with pre-defined depth $\bar{d}$ based on the lower and upper bounds of the prediction range, the probability distribution is estimated by applying the learnable transformation $\Phi(\cdot) : \mathbb{R}^C \rightarrow \mathbb{R}^D$ to the 2D feature vector $h[x] \in \mathbb{R}^C$, followed by the softmax operation over $D$ dimensions:

$$\tilde{d}[x] = \sigma(\Phi(h[x])), \tag{3}$$

where $\sigma$ denotes the softmax operation and $h$ the features obtained after fusing the encodings of image and sparse depth inputs. The vector output $\tilde{d}[x]$ indicates the probability distribution of a feature vector at location $x$ over the discretized $D$ depth planes. In contrast to 2D backprojection that produces sparse 3D samples, our broadcasting distributes 2D features to voxels across depth planes using the estimated probability distribution $\tilde{d}$ following Eq.(3), yielding a full 3D scene representation from the input view.

The broadcasted 3D features of the input view $t$ can be directly fed to the 3D convolutional decoder to predict the dense depth at $t$, as in conventional depth completion methods. This process is straightforward since the encoded features, including the features from different levels of skip connections, are aligned with the same viewpoint as the input view.

### 3.3 LEARNING FROM OCCLUDED REGIONS

The main challenge lies in learning to predict the adjacent view $\tau$ from the encoded features of the input view $t$. To address the difference in perspectives of $t$ and $\tau$, we first warp the view from $t$ to $\tau$ and then complete the empty regions that were occluded in $t$ but are visible in $\tau$. Learning to complete these empty regions in $\tau$ results in an inductive bias.

**(ii) 3D feature warping** is feasible under the assumptions that the 3D scene is static. 3D warping transfers or spatially aligns co-visible features from a input view $t$ to an adjacent view $\tau$. Given 3D features $\mathcal{F}_t$ from the input view $t$ and the relative camera pose $g_{\tau \leftarrow t}$ between views $t$ and $\tau$, and the depth planes with the pre-defined depths $\bar{d}$, the 3D feature warping operation can be denoted as:

$$\mathcal{F}_{\tau \leftarrow t}(x) = \mathcal{F}_t(\pi' g_{t \leftarrow \tau} \bar{X}), \tag{4}$$

where $\bar{X}$ are homogeneous 3D coordinates of the 3D volume that will be projected to $x$ by the canonical projection $\pi'$ assigning features to the nearest voxel location.

**(iii) Predicting the adjacent view feature from the contexts.** The warped 3D feature $\mathcal{F}_{\tau \leftarrow t}$ contains empty voxels (i.e., occluded regions from $t$, presented in $\tau$). To learn an inductive bias to populate the features in the empty voxels, we propose a Contextual eXtrapolation (ConteXt) block as a local context descriptor using $\mathcal{F}_{\tau \leftarrow t}$, along with nearby co-visible regions and their positions, to predict occluded features that appear in the adjacent view.

To derive the context descriptor from nearby *co-visible regions*, we propose a context pooling operation, denoted as $CP(\cdot)$, which aggregates non-empty voxel features through a masked average pool, then upsamples the pooled output by repetition to recover the original feature resolution. Consider each non-overlapping pooling region $R$ of size $k_u \times k_v \times k_w$. The context derived after the proposed context pool can be denoted as:

$$CP(\mathcal{F}_{\tau \leftarrow t})(u, v, w) = \mathcal{U}\left( \sum_{(u,v,w) \in R} \frac{M \odot \mathcal{F}_{\tau \leftarrow t}(u, v, w)}{M(u, v, w) + \epsilon} \right), \tag{5}$$

where $M$ is defined by $M(x) = \mathbf{1}_{\{\mathcal{F}_{\tau \leftarrow t}(x) \neq 0\}}$. $\mathcal{U}$ represents the upsampling operation, repeating the pooled feature within the pooling regions by factors $k_u$, $k_v$, and $k_w$. After context pooling, the context descriptor from Eq. 5 is added to the empty regions of the warped feature $\mathcal{F}_{\tau \leftarrow t}$:

$$\mathcal{F}'_{\tau \leftarrow t} = \mathcal{F}_{\tau \leftarrow t} + \bar{M} \odot CP(\mathcal{F}_{\tau \leftarrow t}), \tag{6}$$

where $\bar{M}$ denotes the complement of the mask $M$, which indicates the positions of originally empty voxels in the warped feature with 1.

To condition the prediction of the adjacent feature on local voxel positions, we encode the 3D sinusoidal positional embedding $\phi$ with $\mathcal{F}'_{\tau \leftarrow t}$. For a single spatial dimension of $u$ the positional embedding $PE_u$ is illustrated as:

$$PE_u(2n) = \sin\left(\frac{u}{\varepsilon^{2n/N}}\right), \; PE_u(2n+1) = \cos\left(\frac{u}{\varepsilon^{2n/N}}\right), \tag{7}$$

where $\varepsilon$ is the frequency constant, and $N$ is the positional embedding dimension. Then, the 3D sinusoidal positional embedding in $(u, v, w)$ is:

$$\phi(u, v, w) = \text{concat}(PE_u, PE_v, PE_w) \in \mathbb{R}^{3N}. \tag{8}$$

Finally, we estimate the adjacent view feature $\hat{\mathcal{F}}_\tau$ from the local contexts via a linear projection layer $g(\cdot)$, which fuses the non-empty region's feature context and the positional context:

$$\hat{\mathcal{F}}_\tau = g(\mathcal{F}'_{\tau \leftarrow t}, \phi, \bar{M}). \tag{9}$$

$\mathcal{F}'_{\tau \leftarrow t}$ denotes the global context descriptor extracted by ConteXt pooling, $\phi$ is a local positional bias and $\bar{M}$ the positions of populated features. While the input view at time $t$ provides features for the co-visible regions, completing the features in the occluded regions requires learning an inductive bias introduced through an additional supervision signal as a loss term.

When predicting the input view depth, the warped feature $\mathcal{F}'_{t \leftarrow t}$ is identical to $\mathcal{F}_t$; whereas the learned positional bias $\phi$ from ConteXt is used to modulate $\mathcal{F}'_{t \leftarrow t}$.

**(iv) Guiding occluded region completion.** Given the unsupervised learning framework, obtaining a supervision signal for the inductive bias through predictions is a natural approach. However, training signals derived from the input data are often limited in quality: they may be sparse (i.e., sparse depth consistency loss) or noisy due to accumulated errors in both estimated camera pose and the predictions (i.e., image reconstruction loss). In this work, we utilize adjacent view features $\mathcal{F}_\tau$ as supervision. The loss of ORCaS serves as an *auxiliary* supervision signal for training ConteXt to learn inductive bias. Specifically, ORCaS leverages synchronized input pairs from the adjacent view to infer its complete features. The proposed loss, $\ell_{\text{ORCaS}}$, enforces consistency between the inferred adjacent view features $\hat{\mathcal{F}}_\tau$ and the encoded adjacent view features, $\mathcal{F}_\tau$. Formally, this loss function that enforces consistency between $\mathcal{F}_\tau$ and $\hat{\mathcal{F}}_\tau$ is denoted as:

$$\ell_{\text{ORCaS-p}} = \sum_x^{\mathcal{X}} ||\hat{\mathcal{F}}_\tau[x] - sg(\mathcal{F}_\tau[x])||_p, \tag{10}$$

Table 1: *Quantitative results on VOID1500 and NYUv2 test sets.* ORCaS outperforms the baselines across all metrics. Compared to (Wu et al., 2024), we improve by an average of 8.91%.

| Method | VOID1500 | | | | NYUv2 | | | |
|---|---|---|---|---|---|---|---|---|
| | MAE ↓ | RMSE ↓ | iMAE ↓ | iRMSE ↓ | MAE ↓ | RMSE ↓ | iMAE ↓ | iRMSE ↓ |
| SS-S2D (Ma et al., 2019) | 178.85 | 243.84 | 80.12 | 107.69 | - | - | - | - |
| DDP (Yang et al., 2019) | 151.86 | 222.36 | 74.59 | 112.36 | - | - | - | - |
| Struct-MDC (Jeon et al., 2022) | 111.33 | 216.50 | - | - | 141.87 | 245.55 | - | - |
| VOICED (Wong et al., 2020) | 85.05 | 169.79 | 48.92 | 104.02 | 127.61 | 228.38 | 28.89 | 54.70 |
| ScaffNet (Wong et al., 2021a) | 59.53 | 119.14 | 35.72 | 68.36 | 117.49 | 199.31 | 24.89 | 44.06 |
| KBNet (Wong & Soatto, 2021) | 39.80 | 95.86 | 21.16 | 49.72 | 105.76 | 197.77 | 21.37 | 42.74 |
| DesNet (Yan et al., 2023) | 37.41 | 93.31 | 19.17 | 45.57 | 103.42 | 188.26 | 21.44 | 38.57 |
| AugUndo (Wu et al., 2024) | 33.32 | 85.67 | 16.61 | 41.24 | 96.73 | 188.70 | 18.95 | 39.18 |
| ORCaS (Ours) | **30.90** | **80.12** | **15.34** | **37.19** | **86.50** | **158.10** | **18.27** | **35.39** |

where $|| \cdot ||_p$ denotes the L-$p$ norm, $sg(\cdot)$ the stop gradient operation, and $\mathcal{X}$ the 3D coordinates.

To predict the adjacent view $\tau$, where relative camera pose $g_{\tau \leftarrow t}$ is non-identity, ConteXt can be applied to the features from both the bottleneck and skip connections to align the viewpoint at view $t$ to the view $\tau$. Importantly, the goal of learning from occlusion is not necessarily to produce high-quality predictions for the adjacent view $\tau$, but to learn an informative inductive bias that enhances the prediction of the input view in $t$, which we can verify by visualizing the adjacent view predictions, as discussed in Sec. 5.

**Predicting depth from 3D features.** Once the 3D features are extracted, they are projected onto a 2D feature space to predict depth for the adjacent view. To do this, we vectorize the 3D features $v$ over the depth planes in each location of the image coordinate $x \in \Omega$, denoted as:

$$r[x] = \text{vec}(\hat{\mathcal{F}}_\tau[x]) \in \mathbb{R}^{C \cdot D}, \tag{11}$$

where $\text{vec}(\cdot)$ denotes the vectorization operation, $C$ is the channel dimension, and $D$ is the number of depth planes. The vectorized feature $v$ is directly fed into the 3D-to-2D projection function $P : \mathbb{R}^{C \cdot D} \to \mathbb{R}^D$, which determines each depth plane's contribution in the 2D features. Next, for each location $x$, these 3D features are weighted by softmax function $\sigma$ over $P(r[x])$. Each element $\sigma(P(r[x]))[d]$ indicates the contribution of the $\mathbb{R}^C$ vector on the $d$-th depth plane. The 3D-to-2D projection to the 2D features reads:

$$\hat{F}_t[x] = \sum_{d=1}^{D} \hat{\mathcal{F}}_t[x][d] \cdot \sigma(P(r[x]))[d], \tag{12}$$

where $\hat{\mathcal{F}}_t[x][d]$ refers to the estimated feature at location $x$ and depth plane $d$. The resulting $\hat{F}_t \in \mathbb{R}^{C \times H \times W}$ is the predicted 2D feature map. Finally, the 2D depth $\hat{d}_t$ is predicted by the output layer $o(\cdot)$, which can be formulated as $\hat{d}_t = o(\hat{F}_t)$.

# 4 EXPERIMENTS

**Results on VOID1500.** We present the quantitative results of ORCaS on VOID1500 compared to unsupervised depth completion baseline models in Tab. 1. By using an auxiliary supervision signal from the adjacent views in ORCaS, we observe an improvement across all evaluation metrics of 62.34% over VOICED (Wong et al., 2020), 45.87% over ScaffNet, 22.87% over KBNet (Wong & Soatto, 2021), 17.68% over DesNet (Yan et al., 2023), and 7.81% over AugUndo (Wu et al., 2024), which is the current state of the art. These gains are primarily driven by a key innovation in ORCaS's design: the inductive bias in the ConteXt block learned by predicting adjacent view features as an auxiliary supervision signal, enabling more accurate depth prediction.

Fig. 2 illustrates the qualitative results on the VOID1500 dataset, emphasizing the strengths of OR-CaS in both homogeneous regions and areas with sharp depth discontinuities. In the left example, ORCaS demonstrates its ability to accurately complete depth in smooth, textureless regions, outperforming the KBNet and AugUndo baselines by leveraging the inductive bias learned from ORCaS

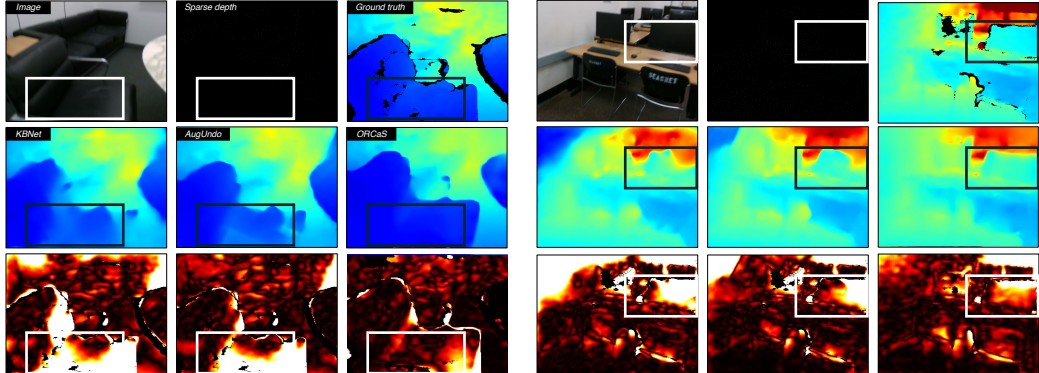

Figure 2: *Qualitative results on VOID1500.* ORCaS improves on homogeneous regions (a leather sofa) in left; and discontinuities (monitors and desks) in right.

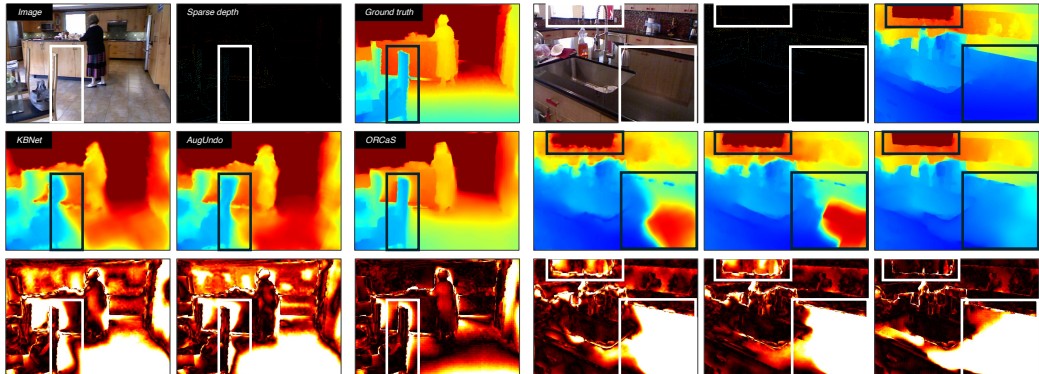

Figure 3: *Qualitative results on NYUv2.* ORCaS improves on homogeneous surfaces (a smoothness of the countertop) in the right; and discontinuities in the left (a chair) and right (windows).

loss. In such large regions with a relatively greater number of sparse depth points, ORCaS is able to learn depth from various views, improving its generalization abilities in large, homogeneous regions compared to baseline methods that learn from a single view. In the right example, ORCaS notably outperforms the baselines in handling depth discontinuities, such as object boundaries and edges. These challenging regions are often problematic due to the sparsity of point clouds, but ORCaS's training strategy—aligning features across views and predicting occluded regions with inductive biases—enables it to predict sharper transitions and more accurate depth in these critical regions.

**Results on NYUv2.** We present the quantitative results of ORCaS on NYUv2 compared to unsupervised depth completion baseline models in Tab. 1. we observe an improvement across all metrics of 33.76% over VOICED (Wong et al., 2020), 23.33% over ScaffNet, 17.49% over KBNet (Wong & Soatto, 2021), and 13.85% over DesNet (Yan et al., 2023), 10.01% over AugUndo (Wu et al., 2024). NYUv2 contains diverse scenes with clutter. Within this challenging scenario, a strong prior is necessary to infer the whole 3D scene. This is precisely the strength of ORCaS, which learns an inductive bias by predicting adjacent views from a single input view. This is evident in the improvement over the state of the art, AugUndo. The qualitative improvements are shown in Fig. 3, where we consistently improve over existing methods as seen by the overall darker (lower) error maps, especially in the homogeneous regions and discontinuous regions, and this may be attributed to the ORCaS's ability to extrapolate using the contextual features.

**Qualitative results of the adjacent view predictions.** While ORCaS's objective is to learn an inductive bias to improve an egocentric depth prediction, we also present the qualitative results on the adjacent view predictions on VOID 1500 test dataset to evaluate the learned inductive bias in Fig. 4. The relative camera poses between the input and the adjacent views are achieved by the pose network finetuned on the test dataset while the depth network being frozen. The evaluation on adjacent views MAE of 79.43, RMSE of 159.82, which outperforms four baselines in Tab. 1. Fig. 4 shows that ORCaS indeed learns an informative inductive bias to predict an adjacent view that aligns well to its scene, depsite not having access.

**Ablation study of components.** We ablate the each component in ORCaS in Tab. 2. The base network (Row 1) is KBNet with a transformer block at the bottleneck. The 3D broadcasting (Row 2) improved it by 2.79%. In Row 5, we choose to warp 3D features (with the depth prediction directly from 2D feature) and warp with ORCaS loss. This is better than the proposed 3D warping without ORCaS loss (Row 3), which is detrimental. Notably, ORCaS loss accounts for 21.6% gain (Rows 4,6) and finally surpasses the

Table 2: *Ablation study on VOID1500 test set*. 2D-3D broadcast denotes 2D-to-3D broadcasting, Warping refers to 2D or 3D warping with relative camera pose, and $\ell_{ORCaS}$ refers to ORCaS loss.

| Method | 2D-to-3D | Warping | $\ell_{ORCaS}$ | MAE ↓ | RMSE ↓ | iMAE ↓ | iRMSE ↓ |
|---|---|---|---|---|---|---|---|
| Base model | | | | 35.31 | 91.32 | 16.61 | 41.13 |
| | ✓ | | | 33.56 | 86.72 | 16.46 | 41.02 |
| | | ✓ | | 52.60 | 125.88 | 28.12 | 66.30 |
| ORCaS | ✓ | ✓ | | 40.52 | 98.87 | 20.61 | 45.51 |
| | | ✓ | ✓ | 36.37 | 90.95 | 17.81 | 43.18 |
| | ✓ | ✓ | ✓ | **30.90** | **80.12** | **15.34** | **37.19** |

Row 1 and 2, which demonstrates the effectiveness of ORCaS loss to learn an inductive bias from completing the occluded feature. This validates the necessity of ORCaS loss to connect the separate components (2D-to-3D broadcast, 3D warping with relative camera pose) to learn from the occlusion as supervision.

**Ablation of the number of depth planes.** We have conducted the ablation study with [2, 4, 8] of the number of depth planes in Tab. 3. ORCaS already achieves the state-of-the-art performance compared to the previous state-of-the-art method (AugUndo) with D=2. As we increase the number of depth planes, the performance further improves.

Table 3: Study on the number of depth planes on VOID.

| | MAE | RMSE | iMAE | iRMSE |
|---|---|---|---|---|
| AugUndo | 33.32 | 85.67 | 16.61 | 41.24 |
| ORCaS (D=2) | 32.73 | 82.47 | 15.96 | 37.60 |
| ORCaS (D=4) | 31.73 | 80.14 | 15.62 | 37.35 |
| ORCaS (D=8) | 30.90 | 80.12 | 15.34 | 37.19 |

**Study on the ConteXt Pooling receptive field.** We tested three different context pooling sizes $(k_u, k_v)$ of [2, 4, 8] in the deepest layer, and multiplied by 2 as the decoder feature resolution is increased. We fixed the pool size of the depth plane to $k_w = 2$, to decouple the effect of the depth plane. Note that the original context pooling size in ORCaS is (4, 4, 2). The result is shown in Tab. 4.

With the ConteXt pooling size of $((k_u, k_v) = (2, 2))$, the performance still improved, yet was limited to 5.67%. This result indicates that the small context pooling size does not have sufficient field of view to inform the completion of empty regions. Our context pooling size of $((k_u, k_v) = (4, 4))$ is the original size that we have used. The context pooling size of $((k_u, k_v) = (8, 8))$ pools the feature coarsely, which smooths the depth plane representation.

**Zero-shot transfer and sensitivity on sparsity.** We evaluate the zero-shot capability of ORCaS trained on VOID1500 to NYUv2 and ScanNet, and conduct a sparsity study on VOID150. The results are presented in Tab. 5. For zero-shot, ORCaS shows an average improvement of 12.1% on NYUv2 and 19.2% on ScanNet, compared to the current state-of-the-art model (AugUndo (Wu et al., 2024)). ORCaS predicting the adjacent views from a single view greatly enhances the generalizability to

Table 4: Ablation study on the ConteXt pooling size $(k_u, k_v, k_w)$.

| $(k_u, k_v, k_w)$ | MAE | RMSE | iMAE | iRMSE |
|---|---|---|---|---|
| (2,2,2) | 32.86 | 83.26 | 16.36 | 38.90 |
| (4,4,2) | **30.90** | 80.12 | **15.34** | **37.19** |
| (8,8,2) | 31.36 | **80.00** | 15.50 | 37.71 |

both NYUv2 and ScanNet. Learning to predict the occluded region requires an inductive bias to the single-view feature to represent not only input views but also adjacent views, where the inductive bias is necessary to infer the shapes populating novel datasets. For VOID150 ($10\times$ reduction), the method demonstrates its superior robustness, with average metric improvements of 31.2% over the state-of-the-art model (AugUndo). Notably, the most significant improvements over the baselines are observed in the RMSE and iRMSE metrics, with 31.9% and 31.4% improvements respectively. Improvements in MAE and iMAE are also substantial, at 30.6% and 30.9%, respectively over AugUndo. The robustness of the proposed approach under varying levels of input point cloud sparsity can be attributed to the inductive bias learned from the occluded regions. These weights for completing the occluded view are shared across both the input and adjacent views' features, $\mathcal{F}_t$ and $\hat{\mathcal{F}}_\tau$, which force the model to learn the underlying scene structure and enable more effective feature reconstruction not only when populating missing regions, but also when the input point clouds are more sparse (yielding also sparser features). Our learning mechanism naturally allows ORCaS to perform completion for extremely sparse point clouds of only 150 point (0.05% of pixels).

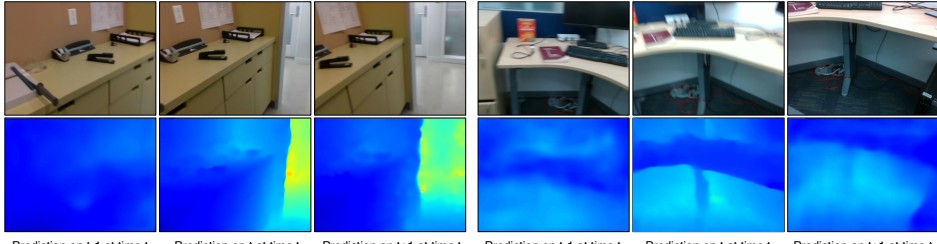

Prediction on t-1 at time t     Prediction on t at time t     Prediction on t+1 at time t     Prediction on t-1 at time t     Prediction on t at time t     Prediction on t+1 at time t

Figure 4: Qualitative results of ORCaS's predicted adjacent views on VOID1500 test.

Table 5: *Zero-shot transfer from VOID1500 to NYUv2 and ScanNet, and Sensitivity study on Sparsity from VOID1500 to VOID150.*

| Method | NYUv2 | | | | ScanNet | | | | VOID150 | | | |
|---|---|---|---|---|---|---|---|---|---|---|---|---|
| | MAE↓ | RMSE↓ | iMAE↓ | iRMSE↓ | MAE↓ | RMSE↓ | iMAE↓ | iRMSE↓ | MAE↓ | RMSE↓ | iMAE↓ | iRMSE↓ |
| VOICED | 2240 | 2427 | 211 | 238 | 1562 | 1764 | 270 | 311 | 209.59 | 329.71 | 130.45 | 229.79 |
| FusionNet | 132.24 | 236.16 | 28.68 | 61.87 | 109.47 | 206.33 | 55.45 | 122.52 | 158.03 | 284.23 | 113.67 | 223.41 |
| KBNet | 138.31 | 257.99 | 25.48 | 51.77 | 103.05 | 217.12 | 36.23 | 76.55 | 149.13 | 306.30 | 70.74 | 136.75 |
| AugUndo | 118.60 | 231.13 | 22.06 | 47.07 | 82.53 | 175.30 | 29.87 | 63.78 | 117.93 | 239.49 | 58.13 | 112.78 |
| ORCaS | **107.68** | **197.48** | **20.05** | **39.85** | **68.86** | **132.93** | **25.23** | **50.77** | **81.89** | **163.06** | **40.16** | **77.38** |

**Feature supervision vs. depth supervision.** We compare the feature supervision vs. the depth supervision to guide the learning of the occluded region in Tab. 6. The supervision with the predicted target depth alone is detrimental to the model performance – this may be due to differences between the predicted depth and the provided sparse depth values, which causes drift. Additionally, there exist multiple combinations of features that yield the same depth, which opens up for ambiguity. Instead, we propose to use the 3D voxel features that are directly extracted from the RGB image and the sparse depth map in the target view. This is higher-dimensional supervision, which is more expressive than depth maps. In the end, we would like to populate the "empty" features consistent with those of other views, so choosing a feature supervision is in fact a more direct supervision than depth maps.

Table 6: Comparison between the depth supervision and feature supervision.

| | MAE | RMSE | iMAE | iRMSE |
|---|---|---|---|---|
| No supervision | 33.56 | 86.72 | 16.46 | 41.02 |
| Depth supervision | 38.81 | 97.38 | 19.35 | 46.28 |
| Feature supervision | **30.90** | **80.12** | **15.34** | **37.19** |

## 5 DISCUSSION AND LIMITATIONS

Although the task of depth completion focuses on estimating depth values for visible surfaces, our work demonstrates that incorporating features in 3D space—predicting beyond visible surfaces—can lead to significant improvements in accuracy. Notably, at inference time, our network operates under the same common setting as *standard depth completion methods*, relying solely on a single RGB image and sparse depth input. We attribute the observed performance gains to the inductive biases that the model learns by mapping features into 3D space and populating it based on constraints derived from multiple views of the same scene during training.

This finding is particularly intriguing because, in principle, depth completion does not necessarily require learning. A simple heuristic approach, such as segmenting an image into local surfaces (e.g., using superpixels) and assigning depth values through sparse depth interpolation, could accomplish the task. However, learning-based methods consistently dominate the benchmarks we test on, raising the question: what additional "hints" do depth completion networks uncover during training beyond simply assigning depth values to surfaces? Our results provide a compelling perspective on this question. By operating in 3D space and forcing the learning of a particular hypothesis to an ill-posed inverse problem, the network acquires a higher-level representation than that resulting from imposing generic priors, e.g., as in 2D depth interpolation. This resulting inductive bias improves the accuracy and generalization of depth estimation as validated by our experiments.

**Limitations.** While ORCaS achieves state-of-the-art performance on depth completion benchmarks, the fundamental reliance on intrinsic calibration may cause sensitivity to noise in these parameters. This dependency could limit ORCaS's applicability in real-world scenarios where camera calibration is error-prone or even unavailable. While from the egocentric view, we observe performance gain, there is no barring that the synthesized occluded view will be free of artifacts.

ACKNOWLEDGMENTS

This paper is supported by the Global Industrial Technology Cooperation Center (GITCC) through a grant agreement with the Korea Institute for Advancement of Technology (KIAT), project number P0028922.

ETHICS STATEMENT

The research in this paper focuses on depth completion for multimodal 3D reconstruction. The intended applications are for beneficial technologies such as autonomous vehicles, robotics, and extended reality (XR). The work was conducted using publicly available datasets and does not involve human subjects or personally identifiable information.

REPRODUCIBILITY STATEMENT

We provide the methodology in Sec. 3 of the main paper and the implementation details in Appendix D. This will be sufficient to reproduce the results. Furthermore, we will release the code and the pretrained weights.

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

## A  FURTHER ORCaS ARCHITECTURE DETAIL

*Upsampling on the prediction.* Unlike the previous methods, ORCaS utilizes dense 3D convolution operations to process the broadcasted feature, which poses a huge computational cost. To address the computational overhead from 3D convolution, we predict the output depth at $1/8$ of the original resolution to mitigate the computational overhead, and the prediction is upsampled to the original resolution by the convex combination of the prediction in location $x$ and its eight neighbors of location $x$.

With an upsampling factor of $\alpha$, we apply a strategy similar to (Teed & Deng, 2020). This method predicts an upsampling mask of dimensions $\alpha \times \alpha \times 3 \times 3$ from 2D feature $F_t[x]$. The upsampling process refines the location $x$ by $\alpha \times \alpha$ using a weighted combination of predictions around $x$, incorporating the eight neighboring locations to enhance accuracy.

## B  DATASETS

We evaluate the proposed Occlusion Completion Network (ORCaS) with the two unsupervised depth completion benchmarks, VOID (Wong et al., 2020) and NYUv2 (Nathan Silberman & Fergus, 2012).

**VOID** (Wong et al., 2020) consists of synchronized RGB images and sparse depth maps with $640 \times 480$ resolution. We use VOID-1500, 104 and 8 sequences for training and testing with varying camera motion and $\approx 1500$ points of a sparse point cloud per instance. The testing set comprises 800 frames. We follow the evaluation protocol of (Wong et al., 2020), where the output depth is assessed against the ground truth points within the range between 0.2 and 5.0 meters. For computational efficiency, two adjacent views are sampled: the frames 10 steps forward and backward from the current frame, ensuring co-visible points between them. We utilize the same number of datapoints for occlusion.

**NYUv2** (Nathan Silberman & Fergus, 2012) consists of 372K synchronized RGB images and sparse point clouds for 464 indoor scenes, with $640 \times 480$ resolution. The training and testing split consists of 249 and 215 scenes, respectively. Following the evaluation protocol in (Wong et al., 2020), ORCaS is evaluated on the test set of 654 images with the $\approx 1500$ points from the depth map sampled by Harris corner detector (Harris et al., 1988) to generate the sparse depth produced by SLAM/VIO (Wong et al., 2020) where output depth is evaluated where ground truth exists between 0.2 and 5.0 meters. For the same as VOID, the adjacent views are sampled 10 frames before and after the current frame. The two adjacent views are selected based on the availability of sparse depth input, where the instances with the adjacent view's input data are discarded. After processing the dataset, 400k samples are utilized to train ORCaS, whereas the other models are trained with 409k samples.

## C  COMPUTATIONAL COST AND INFERENCE SPEED

On VOID1500 with an input size of $640 \times 480$, during inference, ORCaS takes 17.5ms per image (57 FPS). As a reference, KBNet takes 8.6ms per image (115 FPS). Both surpass the real-time threshold of 30 FPS on an Nvidia RTX 3080 GPU. The trade-off is that ORCaS performs 8.91% better than AugUndo over the whole metrics on two depth completion benchmarks, VOID1500 and NYUv2. ORCaS has 24.9M parameters compared to KBNet's 6.96M. However, this amounts to only a 0.34GB difference in GPU memory usage for inference (ORCaS takes 2.35GB memory, KBNet takes 2.01GB), which can easily be handled by commercial GPUs. Also, to validate the baseline with similar number of parameters, we conduct the experiment with a state-of-the-art model (AugUndo (Wu et al., 2024)) with doubled channel size, which amounts to 28.3M parameters. The result shows that even with 12% fewer parameters, ORCas improves 5.16% in MAE, and 6.00% in RMSE on VOID1500.

Table 7: Evaluation of adjacent views predicted by ORCaS on the VOID 1500 test set.

| Method | MAE ↓ | RMSE ↓ | iMAE ↓ | iRMSE ↓ |
|---|---|---|---|---|
| VOICED | 85.05 | 169.79 | 48.92 | 104.02 |
| ORCaS-adj-test | 79.43 | 159.82 | 62.01 | 91.21 |

Table 8: Comparison of the VOID1500 test result to a state-of-the-art method, AugUndo×2.

| Method | # Param | MAE ↓ | RMSE ↓ | iMAE ↓ | iRMSE ↓ |
|---|---|---|---|---|---|
| AugUndo (×2) | 28.3M | 32.58 | 85.24 | 16.01 | 40.19 |
| ORCaS | 24.9M | 30.90 | 80.12 | 15.34 | 37.19 |

## D    IMPLEMENTATION DETAILS

**ORCaS training.** We implemented our method based on the open-sourced code in (Wong & Soatto, 2021) in Pytorch. ORCaS model is optimized by Adam (Kingma & Ba, 2015) with $\beta_1 = 0.9$ and $\beta_2 = 0.999$. For VOID, we used a batch size of 12, with a random crop size of $416 \times 512$. We trained ORCaS for 40 epochs with the initial learning rate of $5 \times 10^{-5}$ for 20 epochs and $2 \times 10^{-5}$ for 20 epochs. We utilized the number of depth planes of $D = 8$. For NYUv2, a batch size of 12 and a random crop size of $416 \times 512$ has been utilized. The sparse depth samples are processed following the open-sourced code in (Wong & Soatto, 2021), which generates a total of 409,343 samples. We trained ORCaS for 12 epochs with the initial learning rate of $1 \times 10^{-4}$ for 4 epochs, $5 \times 10^{-5}$ for an epoch, and $2 \times 10^{-5}$ for 2 epochs, and $5 \times 10^{-6}$ for 5 epochs sequentially. We utilized $D = 8$ depth planes. ORCaS followed the augmentation strategy of AugUndo (Wu et al., 2024).

**Details in sampling adjacent frames.** Following (Wong & Soatto, 2021), the adjacent views are sampled from frames 10 before and 10 after the input frame. Given that VOID and NYUv2 have approximately 30 FPS frame rate, the forward and backward adjacent views are ≈0.33 seconds off from the input view. Note that the adjacent view is generated from the input frame by warping with the input camera pose and inferring the adjacent features. As discussed in the main paper, the adjacent view prediction is not free from artifacts. While our training method affords us the capability of predicting depth maps of different views using inputs only from a single input view, the inductive bias learned through ORCaS also improves generalization to unseen datasets and robustness to various input point cloud sparsity levels.

## E    ADDITIONAL KITTI EXPERIMENTS

The quantitative result of ORCaS on the KITTI depth completion test set is shown in Tab. 9. We observe an improvement across all metrics of VOICED by 19.39%, FusionNet by 13.32%, and AugUndo by 3.36%, where we constantly improve over all metrics. The improvement can be attributed to the key strength of ORCaS: its ability to learn an inductive bias by predicting occluded adjacent views and their relative camera poses from a single input.

Table 9: *Quantitative result on the KITTI DC test set.* ORCaS outperforms the previous SOTA unsupervised depth completion method by 3.44% across all metrics.

| | KITTI DC | | | |
|---|---|---|---|---|
| Method | MAE ↓ | RMSE ↓ | iMAE ↓ | iRMSE ↓ |
| VOICED | 318.59 | 1213.60 | 1.30 | 3.72 |
| FusionNet | 285.55 | 1174.47 | 1.20 | 3.45 |
| AugUndo | 256.37 | 1114.53 | 1.01 | 3.13 |
| ORCaS | 253.17 | 1053.34 | 1.01 | 2.92 |

## F    EVALUATION METRICS

The evaluation metrics used for unsupervised depth completion benchmarks are defined in Tab. 10. The depth completion models are evaluated with Mean Absolute Error (MAE), Root Mean Squared Error (RMSE), inverse Mean Absolute Error (iMAE), and inverse Root Mean Squared Error (iRMSE).

## G    FURTHER DISCUSSIONS

**Sensitivity Study on Sparsity.** In Table 11, we present a sensitivity analysis on the impact of input point cloud sparsity. The VOID500 dataset contains approximately 500 sparse input point clouds.

For **VOID500** (upper table, $3\times$ reduction), the proposed method achieves average improvements across all metrics of 20.3%, 28.5%, 32.1%, 48.4%, and 57.7% compared to AugUndo, DesNet,

Table 10: *Error metrics for depth completion.* $d$ denotes ground truth, and the prediction $\hat{d}$ is evaluated where $d$ values are available for a given image.

| Metric | Definition |
|--------|-----------|
| MAE | $\frac{1}{|\Omega|} \sum_{x \in \Omega} |\hat{d}(x) - d(x)|$ |
| RMSE | $\left( \frac{1}{|\Omega|} \sum_{x \in \Omega} |\hat{d}(x) - d(x)|^2 \right)^{1/2}$ |
| iMAE | $\frac{1}{|\Omega|} \sum_{x \in \Omega} |1/\hat{d}(x) - 1/d(x)|$ |
| iRMSE | $\left( \frac{1}{|\Omega|} \sum_{x \in \Omega} |1/\hat{d}(x) - 1/d(x)|^2 \right)^{1/2}$ |

Table 11: *Quantitative results on VOID500.* The depth completion models are trained on VOID1500 and are tested on VOID500 with different input point cloud sparsity, ORCaS shows average improvement over the baselines of 37.4% on VOID500 across every metric under varying sparsity.

|  | VOID500 | | | |
|--------|-------|-------|-------|--------|
| Method | MAE↓ | RMSE↓ | iMAE↓ | iRMSE↓ |
| VOICED | 137.01 | 235.80 | 71.36 | 130.63 |
| FusionNet | 97.73 | 194.32 | 58.65 | 122.95 |
| KBNet | 78.44 | 178.17 | 37.56 | 83.43 |
| DesNet | 74.89 | 170.32 | 35.62 | 78.30 |
| AugUndo | 66.97 | 151.55 | 31.63 | 71.90 |
| ORCaS | **54.05** | **119.69** | **25.69** | **56.04** |

Table 12: Sensitivity study on camera calibration noise. The calibration noise of 10% and 30% are assumed and evaluated on VOID1500 test set.

|  | ORCaS, ±10% noise, VOID | | | | ORCaS, ±30% noise, VOID | | | |
|------|-------|-------|-------|-------|-------|-------|-------|-------|
| $f$ | 31.52 | 80.62 | 16.10 | 38.38 | 33.92 | 82.37 | 16.58 | 42.89 |
| $c$ | 30.94 | 80.19 | 15.37 | 37.25 | 31.18 | 80.61 | 15.56 | 37.59 |
| $f, c$ | 31.64 | 80.75 | 16.23 | 38.59 | 36.71 | 85.49 | 17.60 | 47.83 |

KBNet, FusionNet, and VOICED, respectively. Notably, the most significant improvements are observed in the inverse metrics, with 39.7% and 39.0% improvements in iMAE and iRMSE, highlighting enhanced performance on closer surfaces. Improvements in MAE and RMSE are also substantial, at 36.7% and 34.2%, respectively. The robustness of the proposed approach under varying levels of input point cloud sparsity can be attributed to its inductive bias learned by predicting occluded region's features. This weight for predicting the occluded view is shared across the features of both the input view and adjacent view, $\mathcal{F}_t$ and $\hat{\mathcal{F}}_\tau$, which forces the model to learn the underlying scene geometry and enables it to effectively reconstruct features and populate sparse regions. Our learning mechanism naturally allows ORCaS to perform completion for extremely sparse point clouds of only 500 point (0.16% of the image space) as illustrated in the bottom section of Tab. 11.

**Sensitivity Study on Camera Calibration Noise.** We assume noise in focal length ($f$) and principal point offset ($c_x, c_y$). We scale the intrinsics by $\{\pm 10\%, \pm 30\%\}$ to simulate calibration error. Tab. 12 shows ORCaS is tolerant of noise up to 30%, which is beyond typical calibration error (e.g., $\approx 0.6$-1.1% using [c]), where performance begins to degrade.

**Visualization on the depth plane probability.** We provide the visualization of the predicted depth plane in Fig. 5. The visualized map presents the predicted depth plane with the highest probability.

**Robustness to Dynamic Objects.** Our training data includes dynamic objects, primarily humans. Their movement can create inconsistencies when features are warped under a static-world assumption. However, ConteXt is trained to correct these artifacts. When an object moves, causing its

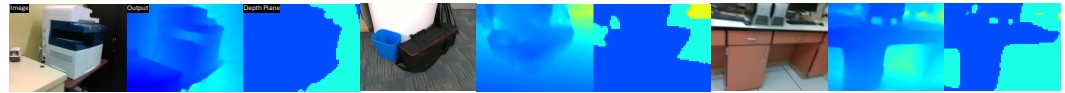

Figure 5: The visualization of depth planes.

Table 13: Study on the effect of the moving object masking during training.

| Setting | VOID | | | |
| | MAE ↓ | RMSE ↓ | iMAE ↓ | iRMSE ↓ |
| --- | --- | --- | --- | --- |
| ORCaS | 30.90 | 80.12 | 15.34 | 37.19 |
| w/o moving | 30.84 | 80.34 | 15.20 | 37.31 |

features to be warped to an incorrect position, ConteXt modulates them to match the object's actual appearance as observed in the adjacent view. An alternative explored by existing work is masking moving objects, which we tested in Tab. 13. The difference is marginal.

## H  FUTURE WORKS

While this paper exploits occlusions of the input view, which is a generic supervision signal, our focus is restricted to depth completion (Lin et al., 2022; Hu et al., 2021; Park et al., 2020; Yang et al., 2019; Wong et al., 2020; 2021b;a; Wong & Soatto, 2021; Zhu et al., 2021; Wu et al., 2024; Park et al., 2024; Park & Jeon, 2024; Chung et al., 2025; Zuo et al., 2025; Rim et al., 2025; Jeong et al., 2025). We foresee our work to be more broadly applicable. The core component of our approach, learning inductive bias through feature completion, can used to learn universal geometric representations with tasks such as optical flow (Aleotti et al., 2020; Lao & Sundaramoorthi, 2017; 2018; Lao et al., 2024a; Sun et al., 2018; Teed & Deng, 2020; Yang & Soatto, 2018; Zhang et al., 2024c;b), monocular depth prediction (Fei et al., 2019; Godard et al., 2019; Gangopadhyay et al., 2025; Lao et al., 2024c; Poggi et al., 2022; Ranftl et al., 2021; Watson et al., 2019; Wong & Soatto, 2019; Upadhyay et al., 2023), visual-language monocular depth estimation (Zeng et al., 2024b;a; 2026) and multi-view stereo (Chen et al., 2019a; Duan & Wong, 2026; Gu et al., 2020; Yao et al., 2018; 2019; Wang et al., 2021; 2024; Zhang et al., 2024a; Wang et al., 2026; 2025b). It is also applicable towards general 2D He et al. (2022); Lao et al. (2024b); Park et al. (2026) and 3D representation learning Chen et al. (2025; 2026). We hope our findings contribute to further exploration of learning 3D scene representations.

