# OpenReview forum: "ORCaS: Unsupervised Depth Completion via Occluded Region Completion as Supervision"
_ICLR.cc/2026/Conference — ICLR 2026 Poster_

### Official Review · Reviewer_ewac · 2025-10-26

**Soundness:** 2
**Presentation:** 3
**Contribution:** 3
**Rating:** 6
**Confidence:** 4

**Summary:**

This paper proposes an unsupervised depth estimation method that augments depth estimation from the target view by introducing additional supervision from the source view. The main assumption is that by using features from the target view to estimate the depth of the source view, the target features can learn to model unseen structures, thereby regularizing the shape of visible structures. Experiments demonstrate notable improvements over previous methods on the VOID1500 and NYUv2 datasets.

**Strengths:**

1. The paper is mostly clearly written and includes proper illustrations.
2. Although reconstructing occluded 3D geometry is not a new concept, applying this idea to unsupervised depth completion is novel and interesting.
3. The proposed model achieves favorable improvements over existing methods.

**Weaknesses:**

1. Rationale of unseen geometry learning:
   The rationale for using improved occluded geometry to enhance visible geometry is not clearly validated. While the authors claim effectiveness in Lines 60–64, the argument remains conceptual without concrete evidence. Since the model itself does not explicitly learn a “3D shape” of objects (L61), it is unclear whether it truly reduces reliance on input point density. Moreover, although the method claims that learning unseen geometry helps improve visible geometry, there are no quantitative or qualitative comparisons in the unseen regions.


2. Lack of ablation experiments:
   - (a) Depth vs. feature supervision: It is unclear why the authors use feature-based supervision instead of depth-based supervision for adjacent views. Depth supervision would be a more direct and intuitive approach and would enable quantitative comparisons in occluded regions to justify the design choice. If depth supervision performs poorly, an explanation should be provided.
   - (b) Computational analysis: The authors mention in L403 that the base model is KBNet with a transformer block. The added transformer head appears to contribute a significant performance gain (MAE: 39.8 → 35.3). This raises the question of whether the improvement is partly due to increased model capacity. A comprehensive comparison of #parameters, GFLOPs, and GPU memory usage among the proposed method, KBNet, and AugUndo is necessary.
   - (c) ConteXt module ablation: Although the authors ablate 2D and 3D representations, they do not ablate the ConteXt module under the 3D representation, nor do they analyze the effect of its hyperparameters $(k_u, k_v, k_w)$. Since this module essentially performs feature pooling, it is important to evaluate how much it contributes to the final performance.


3. Unclear writing and typos:
   - (a) In L241, the authors introduce $\bar{d}$ but do not explain how $\bar{X}$ is derived from $\bar{d}$.
   - (b) In L409–L411, the sentence “(Row 5) This is worse than the proposed 3D warping without ORCaS loss (Row 3)” is inconsistent with the reported results, as Row 5 actually performs better than Row 3.


4. Missing related work on occluded scene reconstruction:
   The following works should be cited and discussed for completeness:
   [1] *Peeking Behind Objects: Layered Depth Prediction from a Single Image*
   [2] *Layer-Structured 3D Scene Inference via View Synthesis*
   [3] *Behind the Scenes: Density Fields for Single-View Reconstruction*
   [4] *Know Your Neighbors: Improving Single-View Reconstruction via Spatial Vision-Language Reasoning*
   [5] *Directed Ray Distance Functions (DRDF) for 3D Scene Reconstruction*
   [6] *X-Ray: A Sequential 3D Representation for Generation*
   [7] *LaRI: Layered Ray Intersections for Single-View 3D Geometric Reasoning*
   [8] *RaySt3R: Predicting Novel Depth Maps for Zero-Shot Object Completion*

**Questions:**

The following experiments and analyses are recommended for the revised version:

1. Replace feature-based supervision with depth-based supervision for adjacent views in the loss function. Analyze whether the predicted depths beyond the visible regions of the target view improve in the source view.
2. Under the 3D representation, ablate the ConteXt module and its hyperparameters $(k_u, k_v, k_w)$.
3. Provide a computational comparison (including #parameters, GFLOPs, and GPU memory) among the proposed method, KBNet, and AugUndo.

---

> ### Author Response · Authors · 2025-11-21
> **Response to Reviewer ewac. (Part1)**
>
> Dear Reviewer ewac,
>
> Thank you for your time and effort on this review and your constructive suggestions to improve our paper. We appreciate your acknowledgment of our paper that (1) applying the idea of occluded region in 3D to unsupervised depth completion is novel and interesting, (2) the proposed model achieves favorable improvements over existing methods, (3) our manuscript is well-written.
>
> We provide further discussions on your questions as follows:
>
> **W1. Rationale of unseen geometry learning:
> The rationale for using improved occluded geometry to enhance visible geometry is not clearly validated. While the authors claim effectiveness in Lines 60–64, the argument remains conceptual without concrete evidence. Since the model itself does not explicitly learn a “3D shape” of objects (L61), it is unclear whether it truly reduces reliance on input point density. Moreover, although the method claims that learning unseen geometry helps improve visible geometry, there are no quantitative or qualitative comparisons in the unseen regions.**
>
>
> We do validate this claim in Tab. 4 in the Supp Mat., where we evaluated on unobserved images in the VOID1500 testing set. This is done by providing an adjacent frame to the image to be evaluated as the input to the depth network along with the relative pose from the input frame to the image to be evaluated. This pose is estimated by the pose network. We show that the predicted depth of unseen views (by querying with estimated pose) shows comparable performance as competing methods (e.g., VOICED) that are given the views as input. We also provided qualitative evidence in Fig. 4 of the main text to show that ORCaS does learn the occluded geometry by predicting unseen views. This validates our hypothesis of learning occluded geometry.
>
> Tab. 3 of the main text shows that for a lower density of the sparse depth, ORCaS performs better than the baselines on VOID150. We also provide evidence of this in Tab. 8 in Supp. Mat, which evaluates on VOID500 comprised of another density level. While we do not explicitly learn the 3D shape of objects, e.g., with object CAD models, we do backproject features to 3D (voxels) and decode them to a reconstruction as a 2.5D range map, which can be seen as a proxy of 3D shape.
>
> **W2a. Lack of ablation experiments:
> (a) Depth vs. feature supervision: It is unclear why the authors use feature-based supervision instead of depth-based supervision for adjacent views. Depth supervision would be a more direct and intuitive approach and would enable quantitative comparisons in occluded regions to justify the design choice. If depth supervision performs poorly, an explanation should be provided.**
>
> Yes, we did try the depth supervision before. The supervision with the predicted target depth alone is detrimental to the model performance -- this may be due to differences between the predicted depth and the provided sparse depth values, which causes drift. Additionally, there exist multiple combinations of features that yield the same depth, which opens up for ambiguity. Instead, we propose to use the 3D voxel features that are directly extracted from the RGB image and the sparse depth map in the target view. This is higher-dimensional supervision, which is more expressive than depth maps. In the end, we would like to populate the "empty" features consistent with those of other views, so choosing feature-based supervision is in fact a more direct supervision signal than depth maps.
>
> The Table below shows the results of without any supervision, the depth supervision, and the feature supervision.
> |       Supervision      |  MAE  |  RMSE  |  iMAE  |  iRMSE  |
> |:------------:|:------:|:------:|:------:|:------:|
> | No supervision | 33.56 | 86.72 | 16.46 | 41.02 |
> | Depth supervision | 38.81 | 97.38 |19.35 | 46.28 |
> | Feature supervision | 30.90 | 80.12 |15.34 | 37.19 |

---

> ### Author Response · Authors · 2025-11-21
> **Response to Reviewer ewac. (Part2)**
>
> **W2b. Computational analysis: The authors mention in L403 that the base model is KBNet with a transformer block. The added transformer head appears to contribute a significant performance gain (MAE: $39.8 \rightarrow 35.3$). This raises the question of whether the improvement is partly due to increased model capacity. A comprehensive comparison of num. parameters, GFLOPs, and GPU memory usage among the proposed method, KBNet, and AugUndo is necessary.**
>
> Model capacity plays a small role. Tab. 5 in Supp. Mat. shows that, with increased capacity, AugUndo with KBNet cannot reach our performance. While the transformer block did help improve, this is a minor innovation that we did not find worthy of a major contribution.
> Nonetheless, it does contribute to performance gain (11.5\%). Yet the improvement of our major contributions improves over that from 35.3 to 30.9 (12.5\%); an improvement on top of already high gain is difficult, and the higher percentage gain through ORCaS demonstrates the effectiveness of our method.
>
> In terms of computations, we do not lose out in speed and only incur 0.3GB more memory for inference: Our proposed model improves KBNet by 20.18\% and AugUndo by 8.91\%, where both models use the KBNet architecture; the trade-off of ORCaS to KBNet is 24M vs. 7M parameters, 45.12GFLOP vs. 24.53GFLOP. This amounts to 2.3GB GPU memory and 17ms per image (58 FPS) for ORCaS and 2GB GPU memory and 8.6ms per image (115 FPS) for KBNet and AugUndo for the inference. Both were benchmarked on an RTX 3080 GPU. Yet, we're still running in real-time and have no latency due to inference speed, since the average camera input frequency is 30 FPS, which we exceed with 58 FPS.
>
> **W3. Unclear writing and typos:
> (a) In L241, the authors introduce $\bar{d}$ but do not explain how $\bar{X}$ is derived from$ \bar{d}$.
> (b) In L409–L411, the sentence “(Row 5) This is worse than the proposed 3D warping without ORCaS loss (Row 3)” is inconsistent with the reported results, as Row 5 actually performs better than Row 3.**
>
> (a) The $\bar{X}$ is derived by distributing the feature $X$ in grid $(u,v)$ over $D$ depth planes.
> This is done by multiplying the feature with the predicted softmax probability distribution over depth planes, where the probability distribution is predicted by the linear layer with output dimension of $D$. Each depth plane has the pre-assigned depth $\bar{d}$, and we assume the voxel position of $\bar{X}$ indicates the 3D coordinate where the feature point is in.
>
> (b) Sorry, this was a typo. The description should read: ``In Row 5, we choose to warp 3D features (with the depth prediction directly from 2D feature) and warp with ORCaS loss. This is better than the proposed 3D warping without ORCaS loss (Row 3), which is detrimental."
>
> We will fix these in the next revision.
>
> **W4. Missing related work on occluded scene reconstruction**
>
> We will add a paragraph in the Related Works section to discuss the citations accordingly in the next revision.
>
> **Q1a. Replace feature-based supervision with depth-based supervision for adjacent views in the loss function.**
>
> The Table above (under W2a) is the result that we replaced the feature-based supervision with the depth-based supervision. Using depth supervision is detrimental, which is worse than no direct supervision on the features learned.
>
> **Q1b. Analyze whether the predicted depths beyond the visible regions of the target view improve in the source view.**
>
> We do validate this claim in Tab. 4 in the Supp Mat., where we evaluated on unobserved adjacent views in the VOID1500 testing set. This is done by providing an adjacent frame to the image to be evaluated as the input to the depth network along with the relative pose from the input frame to the image to be evaluated. This pose is estimated by the pose network. We show that the predicted depth of unseen views (by querying with estimated pose) shows comparable performance as competing methods (e.g., VOICED) that are given the views as input. We also provided qualitative evidence in Fig. 4 of the main text to show that ORCaS does learn the occluded geometry by predicting unseen views. This validates our hypothesis of learning occluded geometry.

---

> ### Author Response · Authors · 2025-11-22
> **Response to Reviewer ewac. (Part3)**
>
> **Q2. Under the 3D representation, ablate the ConteXt module and its hyperparameters.**
>
> Without the ConteXt module (and without ORCaS loss), the model shows $5.34\%$ of degradation.
> We tested three different context pooling sizes ($k_u, k_v$) of [2, 4, 8] in the deepest layer, and multiplied by 2 as the decoder feature resolution is increased. We fixed the pool size of the depth plane to $k_w=2$, to decouple the effect of the depth plane. Note that the original context pooling size in ORCaS is (4, 4, 2).
>
> With the ConteXt pooling size of ($(k_u, k_v)=(2,2)$), the performance still improved, yet was limited to $5.67\%$. This result indicates that the small context pooling size does not have sufficient field of view to inform the completion of empty regions.
> Our context pooling size of ($(k_u, k_v)=(4,4)$) is the original size that we have used.
> The context pooling size of ($(k_u, k_v)=(8,8)$) pools the feature coarsely, which smooths the depth plane representation.
>
>
> |       $(k_u, k_v, k_w)$        |  MAE  |  RMSE  |  iMAE  |  iRMSE  |
> |:------------:|:------:|:------:|:------:|:------:|
> | (2,2,2) | 32.86 | 83.26 | 16.36 | 38.90 |
> | (4,4,2) | 30.90 | 80.12 | 15.34 | 37.19 |
> | (8,8,2) | 31.36 | 80.00 | 15.50 | 37.71 |
>
>
>
> **Q3. Provide a computational comparison (including \# parameters, GFLOPs, and GPU memory) among the proposed method, KBNet, and AugUndo.**
>
> Please see above (W2b).

---

> ### Comment · Reviewer_ewac · 2025-11-23
>
> I appreciate the new results and analyses provided by the authors. I appreciate the results that validate the improvement in occluded scene reconstruction, as well as the real-time performance of the proposed approach. I will keep my score and vote for acceptance for this paper.

---

### Official Review · Reviewer_k3VB · 2025-10-29

**Soundness:** 3
**Presentation:** 3
**Contribution:** 3
**Rating:** 6
**Confidence:** 5

**Summary:**

This paper presents ORCaS, a new unsupervised depth completion method. The core idea is to treat occluded regions as a source of self-supervision to learn a stronger, 3D-aware inductive bias for reconstructing dense depth maps from sparse depth inputs and RGB images. Concretely, ORCaS broadcasts 2D features into a 3D voxel grid, performs rigid 3D warping using relative poses, predicts “empty” regions in adjacent views, employs a ConteXt block to extrapolate local contextual features, and introduces a new ORCaS loss that learns inductive priors from these occluded regions. Experiments on VOID1500, NYUv2, and KITTI demonstrate state-of-the-art performance.

**Strengths:**

1. Clear motivation. ORCaS introduces the novel concept of occluded region completion as a supervision signal for unsupervised depth learning. By explicitly predicting unseen regions, the method enforces the model to learn a 3D-structure-aware inductive bias that goes beyond traditional visible-region reconstruction.

2. Well-structured design. The architecture is modular, interpretable, and easily integrable with existing unsupervised depth completion frameworks. It can serve as a plug-and-play component for similar tasks.

3. Comprehensive validation. Extensive experiments across VOID1500, NYUv2, and KITTI datasets demonstrate consistent and significant performance gains. Ablation studies and transfer experiments further support the effectiveness of each design choice.

4. Strong generalization. By learning to predict occluded regions, ORCaS acquires a geometry-aware prior that improves cross-dataset transfer and remains robust even with extremely sparse depth.

**Weaknesses:**

1. Limited theoretical explanation of ORCaS loss. While the paper empirically demonstrates the effectiveness of occlusion-based supervision, it lacks a deeper theoretical analysis explaining why predicting unobserved regions improves the learned representation for visible depth estimation.

2. Dependency on accurate camera calibration. The method relies on precise camera intrinsics and relative poses. Although this limitation is acknowledged, the paper does not include ablation or robustness studies to quantify sensitivity to calibration noise.

3. Outdated related work. The literature review mainly covers works up to 2023. It is recommended to expand this section to include up‐to‐date publications, such as:

[1] Distilling Monocular Foundation Model for Fine-grained Depth Completion. CVPR 2025.

[2] Completion as Enhancement: A Degradation-Aware Selective Image Guided Network for Depth Completion. CVPR 2025.

[3] OMNI-DC: Highly Robust Depth Completion with Multiresolution Depth Integration. ICCV 2025.

[4] PacGDC: Label-Efficient Generalizable Depth Completion with Projection Ambiguity and Consistency. ICCV 2025.

[5] Tri-Perspective View Decomposition for Geometry-Aware Depth Completion. CVPR 2024.

**Questions:**

I am willing to increase the rating if those weaknesses can be addressed in the rebuttal stage, thanks.

---

> ### Author Response · Authors · 2025-11-21
> **Response to Reviewer k3VB. (Part1)**
>
> Dear Reviewer k3VB,
>
> We appreciate your time and effort on this review and your constructive suggestions to improve our manuscript. We appreciate the acknowledgment that the paper is well-written, the motivation is clear, and the method generalizes well.
>
> **W1. Limited theoretical explanation of ORCaS loss. While the paper empirically demonstrates the effectiveness of occlusion-based supervision, it lacks a deeper theoretical analysis explaining why predicting unobserved regions improves the learned representation for visible depth estimation.**
>
> There exist infinitely many 3D scenes that may be present "behind" occluders, e.g. occluded regions. Naturally, like any ill-posed problem as such, one must make assumptions, regularize, or bias predictions to a subset of plausible or preferred 3D scenes or solutions. Rather than a generic regularizer that models typical regularities such as local smoothness and connectivity, we choose one that leverages the consistency in the projection of objects onto the image plane. Our proposed method backprojects features (to 3D) and completes regions (which are composed of objects) such that their projection (both features and depth) matches those in other views. This implicitly enforces the learning of objects in 3D.
>
> Additional evidence of improvements from learning to predict unobserved regions is Masked Autoencoding, where they complete the intensities of images, which also biases transitions of colors or textures corresponding to objects across pixels. On the theoretical side, as the Masked AutoEncoder can be viewed as maximizing the mutual information between the visible and masked (occluded) regions, ORCaS also forces the learning of the inductive bias by the maximization of mutual information through minimizing the reconstruction loss between the adjacent views' features and the features warped and completed from the input view.
>
>
>
> **W2. Dependency on accurate camera calibration. The method relies on precise camera intrinsics and relative poses. Although this limitation is acknowledged, the paper does not include ablation or robustness studies to quantify sensitivity to calibration noise.**
>
> We do show the impact of the camera intrinsics error during inference in Table 9 of the Supp. Mat. and L855-858. We do not assume a precise and accurate relative pose. As the relative pose is predicted by a pose network, and jointly learned with the depth estimator, the pose is expected to be noisy throughout the majority of the optimization process.
>
>
> **W3. Outdated related work. The literature review mainly covers works up to 2023.**
>
> We will add a paragraph in the Related Works section to discuss the citations accordingly in the next revision.

---

> > ### Comment · Reviewer_k3VB · 2025-11-22
> >
> > Thanks for the feedback which has addressed my concerns. I vote for acceptance.

---

### Official Review · Reviewer_gNmQ · 2025-10-31

**Soundness:** 4
**Presentation:** 4
**Contribution:** 4
**Rating:** 6
**Confidence:** 3

**Summary:**

This paper introduces a novel unsupervised framework that learns dense depth estimation from an RGB image and sparse point cloud by explicitly reasoning about occluded 3D regions. Rather than relying solely on photometric reconstruction of co-visible areas, the paper proposes to learn an inductive 3D bias through the auxiliary task of occluded region completion. ORCAS, a method of the paper, first encodes RGB and sparse depth into 2D features, broadcasts them into a discretized 3D volume, and rigidly warps this volume to an adjacent view using relative pose. The ConteXt block then fills in the empty voxels corresponding to occluded regions using nearby 3D context and positional embeddings, while a new ORCaS loss enforces consistency between predicted and real adjacent-view features. This occlusion-aware training significantly improves the performance of depth predictions in an unsupervised setting. Extensive experiments on VOID1500, NYUv2, and ScanNet show that ORCaS achieves state-of-the-art performance, outperforming previous unsupervised methods by up to 8.9% on average, while maintaining real-time inference speed and demonstrating strong robustness to domain shifts, calibration noise, and extremely sparse depth inputs

**Strengths:**

	ORCaS introduces a simple yet novel idea, using occluded region completion as an auxiliary supervision signal for unsupervised depth completion. This reframes depth completion from a purely visible-surface interpolation problem into a 3D reasoning task that requires understanding unseen geometry. By leveraging occlusion as supervision, the method naturally learns a strong inductive bias that encourages consistent 3D representations. This conceptual clarity and originality make the paper both theoretically appealing and practically impactful.

	Across multiple benchmarks (VOID1500, NYUv2, ScanNet), the method consistently achieves state-of-the-art performance, outperforming previous unsupervised methods by up to 8.9% on average.

	The proposed method learns latent features that encode the 3D shape regularities of indoor scenes, independent of texture or lighting. Even though the model is not directly trained for domain transfer, this implicit shape prior helps it perform well in zero-shot transfer and sparse-input settings.

	Authors demonstrate strong robustness to variations in calibration, scene dynamics, and input sparsity. It maintains stable performance even with +-30% synthetic calibration noise and when trained on static assumptions in dynamic environments.

**Weaknesses:**

Major weaknesses are as below:

	Most experiments focus on indoor or small-scale environments (VOID1500, NYUv2, ScanNet). The KITTI Depth Completion results are included only in the appendix, where the improvement over prior work is relatively small (≈3%). This suggests that the learned occlusion-based bias may generalize less effectively to outdoor, long-range, or high-depth-variance settings. A broader evaluation would be necessary to confirm the scalability of the approach beyond indoor domains.

	Although the method is built around the idea of learning from occluded-region completion, the qualitative results do not visually emphasize or analyze regions where occlusion is likely to occur. Figures 2 and 3 mainly show overall depth predictions for relatively frontal or fully visible areas, rather than viewpoints where depth discontinuities, inter-object occlusions, or self-occlusions are pronounced. Without explicitly highlighting or comparing, it is difficult to tell whether the proposed occlusion reasoning truly contributes to the improved depth quality.


Minor comments are as below:

	In the ablation section, the text description around Table 2 incorrectly describes the relative performance between Row 3 and Row 5. The numbers in the table show that Row 5 performs better, but the text argued in the opposite.

	The paper mentions that training is performed “in an alternating fashion” in L91-92, but provides no further explanation or details about what this process entails. There is no description of how the alternation is implemented, what modules are updated in each phase, or why this strategy is necessary.

	A comparison with the baseline model, KBNet, are not presented in table 6. Following the KITTI benchmark performance gap between the proposed method and KBNet is very marginal.

**Questions:**

	Could you elaborate on why the proposed occlusion-completion supervision may generalize less effectively to outdoor environments? Have you tested the method on any additional large-scale or high-depth-variance datasets to evaluate scalability beyond indoor domains?

	Please explain how the alternating training process is scheduled (per batch, per epoch, or per iteration), which parameters are frozen in each phase, and why this two-step optimization was preferred over joint training.

	Could you provide visualizations or case studies focusing specifically on occluded or partially visible areas? How can we confirm that the learned ConteXt block completes unseen regions rather than merely smoothing co-visible surfaces?

	The current setup uses only two adjacent frames for occlusion-aware supervision. Have you explored extending ORCaS to longer temporal windows or multiple adjacent views? Incorporating multi-frame context might improve occlusion stability and reduce dependence on single-pose accuracy. Do you expect the current ConteXt block or ORCaS loss to generalize naturally to that setting?

	It would be interesting to know whether ORCaS could serve as a pretraining stage for other 3D perception tasks such as monocular depth estimation or scene flow. Do you believe the learned occlusion-aware features transfer effectively to other geometry-related tasks?

---

> ### Author Response · Authors · 2025-11-21
> **Response to Reviewer gNmQ. (Part1)**
>
> Dear Reviewer gNmQ,
>
> We appreciate your time and effort on this review and your constructive suggestions to improve our manuscript. We appreciate the acknowledgment that the method is simple yet novel, the motivation is clear, and the method is robust to the calibration noise.
>
> Responses below are marked with Q\# for Question, W\# for Weakness, M\#for Minor Comment.
>
> **W1a. The KITTI Depth Completion results are included only in the appendix, where the improvement over prior work is relatively small ($\approx$3).**
>
> While improvement on KITTI is smaller than that of VOID1500 and NYUv2 (average of 8.91\%), we hypothesize this is due to the saturation of the benchmark. Improvement of each SOTA over the prior art (e.g., AugUndo over KBNet) is only by 1.69\%, and subsequent improvements become more difficult. The fact that our improvement is more than that of prior arts suggests that our method does generalize well to outdoor, long-range, and high-depth-variance settings.
>
>
> **M1. In the ablation section, the text description around Table 2 incorrectly describes the relative performance between Row 3 and Row 5. The numbers in the table show that Row 5 performs better, but the text argues the opposite.**
>
> Sorry, this was a typo. The description should read: ``In Row 5, we choose to warp 3D features (with the depth prediction directly from 2D feature) and warp with ORCaS loss. This is better than the proposed 3D warping without ORCaS loss (Row 3), which is detrimental." We will fix those in the next revision.
>
> **M2. The paper mentions that training is performed “in an alternating fashion” in L91-92, but provides no further explanation or details about what this process entails. There is no description of how the alternation is implemented, what modules are updated in each phase, or why this strategy is necessary. Explain how the alternating training process is scheduled (per batch, per epoch, or per iteration), which parameters are frozen in each phase.**
>
> Alternating optimization between the ConteXt module and the remaining (both depth and pose networks) parameters is performed each training iteration. In the first alternation step of each iteration, we freeze the ConteXt module ($g(\cdot)$) and train the other parameters with the photometric and sparse depth reconstruction losses and a local smoothness regularizer (Eq. 2, main paper). In the other alternation step, we freeze the other parameters and train only ConteXt module $g(\cdot)$ to predict the features on the adjacent views with ORCaS-p loss (Eq. 10, main paper).
>
> **M3. A comparison with the baseline model, KBNet, are not presented in table 6. Following the KITTI benchmark performance gap between the proposed method and KBNet is very marginal.**
>
> Table 6 in the Supp. Mat. reports on the KITTI on validation set. KBNet results (taken from their paper) on the validation set are as follows: MAE=260.44 RMSE=1126.85 iMAE=1.03 iRMSE=3.20. Our method achieves MAE=253.17 RMSE=1053.34 iMAE=1.01 iRMSE=2.92. We improve by an average of 5\% across all metrics, which is nontrivial.
>
> **Q1a. Could you elaborate on why the proposed occlusion-completion supervision may generalize less effectively to outdoor environments?**
>
> We attribute this to the saturation of the benchmark. While we improve by approximately 3 percent, we note that previous improvements have been on the scale of less than 2 percent. Further improvements tend to be more difficult, so in fact, improving by 3 percent demonstrates that our method is effective for outdoor environments.
>
> **Q2. Why this two-step optimization was preferred over joint training?**
>
> It is necessary because the supervision depends on an unobserved adjacent view, which is a chicken-and-egg problem in the case of unsupervised learning of depth due to the lack of ground truth (and also features). Therefore, we must alternate and freeze the parameters, otherwise the training degrades the model performance on the current scene.

---

> ### Author Response · Authors · 2025-11-22
> **Response to Reviewer gNmQ. (Part2)**
>
> **Q3b. How can we confirm that the learned ConteXt block completes unseen regions rather than merely smoothing co-visible surfaces?**
>
> We enforce this behavior during training by explicitly supervising ConteXt with the features from adjacent views, which contain unseen regions. We validate this on the VOID1500 testing set in Fig. 4 in the main text, and we show that indeed ConteXt enables extrapolation of the unseen regions.
> Note that we also quantitatively validate this on the testing set in Tab. 4 in the Supp. Mat, where we show that the predicted depth of unseen views (by querying with estimated pose) shows comparable performance as competing methods (e.g., VOICED) that are given the views as input.
>
> **Q4a. The current setup uses only two adjacent frames for occlusion-aware supervision. Have you explored extending ORCaS to longer temporal windows or multiple adjacent views?**
>
> Yes, we do. The Table below shows that we improve if we were to train with larger intervals (increasing from 10 frames to 20 frames and 30 frames apart), our scores improve on VOID1500.
>
> |       Model        |  MAE  |  RMSE  |  iMAE  |  iRMSE  |
> |:------------:|:------:|:------:|:------:|:------:|
> | AugUndo  |     33.32 | 85.67 | 16.61 | 41.24 |
> | ORCaS (delta_t=10) |  30.90 | 80.12 | 15.34 | 37.19 |
> | ORCaS (delta_t=20) |  30.61 | 80.00 | 15.31 | 37.12 |
> | ORCaS (delta_t=30) |  30.21 | 79.81 | 15.12 | 36.89 |
>
> **Q4b. Incorporating multi-frame context might improve occlusion stability and reduce dependence on single-pose accuracy. Do you expect the current ConteXt block or ORCaS loss to generalize naturally to that setting?**
>
> While this is out of the scope of our work, yes, we do expect ConteXt or ORCaS to generalize to those settings. With multiple frames, pose estimation and depth estimation for a given view naturally improve; yet, occlusions are still present across frames, which is what we exploit as supervision.
>
> **Q5. It would be interesting to know whether ORCaS could serve as a pretraining stage for other 3D perception tasks, such as monocular depth estimation or scene flow. Do you believe the learned occlusion-aware features transfer effectively to other geometry-related tasks?**
>
> While this is also out of the scope of our work, yes, we do expect ConteXt and ORCaS to serve as pretraining to monocular depth (i.e., depth completion without sparse depth) and scene or optical flow as occlusions are still present. One evidence of this is a Masked AutoEncoder (MAE), which artificially simulates occlusion by removing patches from input images. The community has seen success when using MAE as pretraining for subsequent geometric tasks. Our method also follows a similar intuition, but occlusions are naturally induced by motion.

---

> > ### Comment · Reviewer_gNmQ · 2025-11-24
> >
> > Thanks for the detailed explanation. Most of my concerns have been addressed, and I will keep my score to accept this paper.

---

> ### Author Response · Authors · 2025-12-03
> **Response to Reviewer gNmQ (Part 3).**
>
> **W1b. A broader evaluation would be necessary to confirm the scalability of the approach beyond indoor domains.**
>
> To demonstrate that ORCaS can scale beyond indoor domains, we evaluated ORCaS on the outdoor domain (the KITTI DC online benchmark, testing set). Compared to the previous SOTA method (DesNet), we improved by 3.7\% on average over the four metrics (MAE, RMSE, iMAE, iRMSE). The results are shown below:
>
> |       Model        |  MAE  |  RMSE  |  iMAE  |  iRMSE  |
> |:------------:|:------:|:------:|:------:|:------:|
> | DesNet  |   266.24 | 938.45  | 1.13 | 2.95 |
> | ORCaS  |  246.72 | 931.77 | 1.13 | 2.75|

---

### Official Review · Reviewer_pttx · 2025-10-31

**Soundness:** 3
**Presentation:** 3
**Contribution:** 3
**Rating:** 6
**Confidence:** 4

**Summary:**

This paper addresses the self-supervised depth completion task by introducing an auxiliary objective: completing occluded regions of the scene. This auxiliary task serves as a strong inductive bias to guide the learning process for depth completion. Experimental results on the VOID1500 and NYUv2 datasets demonstrate that the proposed approach achieves superior performance compared to previous methods.

**Strengths:**

- The paper is clearly written and well organized.
- The proposed method is sound.
- The proposed method demonstrates superior performance compared to existing approaches on indoor datasets.

**Weaknesses:**

1. The main text includes comparisons only on two indoor datasets. Although KITTI Depth Completion results are reported in the supplementary material, the comparison involves only a limited number of competing methods. Moreover, the performance on the KITTI DC dataset appears inferior to several previous approaches, such as DesNet. It is recommended to provide a more detailed analysis of the results on outdoor datasets to better demonstrate the effectiveness and robustness of the proposed method.

**Questions:**

1. How is the relative camera pose obtained? Is it predicted by a network or derived from ground-truth camera poses?
2. Besides occluded regions, there are areas that do not overlap between two frames. Would these non-overlapping regions affect the depth completion learning process?
3. The difficulty of scene completion is related to the time interval between frames, as a larger interval typically results in more occluded regions. How do you determine an appropriate frame interval to best assist depth completion learning?
4. Could you provide an analysis of the impact of the number of planes used for MPI on the overall performance?

---

> ### Author Response · Authors · 2025-11-21
> **Response to Reviewer pttx. (Part1)**
>
> Dear Reviewer pttx,
>
> Thank you for your time and effort on this review and your constructive suggestions to improve our paper.
>
> Responses below are marked with Q\# for Question and W\# for Weakness.
>
> **Q1. How is the relative camera pose obtained? Is it predicted by a network or derived from ground-truth camera poses?**
>
> During training, the relative camera pose is predicted by the separate pose network, which is trained with the photometric and sparse depth reconstruction losses with a local smoothness regularizer (Eq. 2, from main text).
>
> **Q2. Besides occluded regions, there are areas that do not overlap between the two frames. Would these non-overlapping regions affect the depth completion learning process?**
>
> These non-overlapping regions are precisely what is being exploited as supervision during the learning process. We learn to complete (and predict) these regions by supervising ConteXt to produce features of a target frame from an input source or reference frame. This forces the learning of an inductive bias, which improves generalization. We further validate this in the table below where we increase the time interval (from 10 frames to 20 and 30 frames apart) between adjacent frames so that more regions are non-overlapping between frames. Indeed, our performance improves when more regions are occluded and unseen.
>
> **Q3. How do you determine an appropriate frame interval to best assist depth completion learning?**
>
> We followed the time interval and training protocol set by the KITTI and VOID1500 benchmarks: each KITTI training example is comprised of three images where each image is one frame apart from its previous or next frame, each VOID1500 training example is comprised of three images where each image is ten frames apart. Given that KITTI is 10 FPS, this amounts to 100ms of interval between frames. VOID1500 is 30 FPS, which gives a 333.33ms interval per ten frames. However, the appropriateness is less about time, but more about motion. Like general SfM, so long as there is sufficient parallax and covisible regions, one can utilize photometric reprojection loss as the data term for supervision.
>
> An advantage of our method is that we make use of occluded regions, which existing methods cannot; hence, we hypothesize that training with a larger interval (or frames apart) can help as typically larger intervals correlate with more occluded and non-overlapping regions. The Table below shows that if we were to train with larger intervals (20 frames, 30 frames), our scores improve on VOID1500. However, like SfM and other unsupervised methods, when there are little to no covisible regions across frames (100 frames, 200 frames apart), i.e., no data term due to lack of correspondences, our training collapses.
>
> |     Model         |  MAE  |  RMSE  |  iMAE  |  iRMSE  |
> |:----------------:|:------:|:------:|:------:|:------:|
> |  AugUndo       | 33.32 | 85.67 | 16.61 | 41.24 |
> |  ORCaS (delta_t=10) | 30.90 | 80.12 | 15.34 | 37.19 |
> |  ORCaS (delta_t=20) | 30.61 | 80.00 | 15.31 | 37.12 |
> |  ORCaS (delta_t=30) | 30.21 | 79.81 | 15.12 | 36.89 |
>
> **Q4. Could you provide an analysis of the impact of the number of planes used for MPI on the overall performance?**
>
> We have conducted the ablation study with [2, 4, 8] of the number of depth planes, in the Table below. The result shows the number of depth planes for VOID1500. ORCaS already achieves the state-of-the-art performance compared to the previous state-of-the-art method (AugUndo) with D=2. As we increase the number of depth planes, the performance further improves.
>
> |     Model         |  MAE  |  RMSE  |  iMAE  |  iRMSE  |
> |:----------------:|:------:|:------:|:------:|:------:|
> |  AugUndo       | 33.32 | 85.67 | 16.61 | 41.24 |
> |  ORCaS (D=2) | 32.73 | 82.47 | 15.96 | 37.60 |
> |  ORCaS (D=4) | 31.73 | 80.14 | 15.62 | 37.35 |
> |  ORCaS (D=8) | 30.90 | 80.12 | 15.34 | 37.19 |

---

> > ### Comment · Reviewer_pttx · 2025-11-24
> >
> > Thank the authors for their detailed feedback. My concerns are addressed and I decide to keep my rating of acceptance.

---

> ### Author Response · Authors · 2025-12-03
> **Response to Reviewer pttx. (Part 2)**
>
> **W1. A limited number of competing methods in KITTI benchmark. The performance on the KITTI DC dataset appears inferior to ... DesNet. It is recommended to provide a more detailed analysis of the results on outdoor datasets to better demonstrate the effectiveness and robustness of the proposed method.**
>
> We aimed to compare with the SOTA method (AugUndo) on KITTI DC, but it was only reported on the validation set; hence, Table 6 in the Supp. Mat. also followed the reporting convention of AugUndo, which did not include DesNet. Note: AugUndo reported improvements over DesNet on all other datasets. We also note that DesNet reported results on the KITTI DC online test set and trained on both the training and validation set (see the first two sentences in Section 4.1 of their main paper). We followed their training protocol in the Table below and achieved 3.7\% improvement over DesNet on the testing set.
>
> |       Model        |  MAE  |  RMSE  |  iMAE  |  iRMSE  |
> |:------------:|:------:|:------:|:------:|:------:|
> | DesNet  |   266.24 | 938.45  | 1.13 | 2.95 |
> | ORCaS  |  246.72 | 931.77 | 1.13 | 2.75|

---

### Author Response · Authors · 2025-12-03
**Summary of the Rebuttal Period for Area Chair.**

Dear Handling Area Chair,

We thank the reviewers for their thoughtful feedback, and we appreciate your coordination and support throughout the review process, which has helped us improve the manuscript.

During the rebuttal phase, all reviewers responded positively to our revisions and clarifications. Notably, all four reviewers voted for acceptance. All updates were completed by 21:54 UTC, 24 November 2025, which predates the public disclosure of the recent information leak (widely reported between 14:00 and 16:00 UTC on 27 November 2025).

Below, we summarize our rebuttal efforts, which include:
(1) Additional results to answer the reviewers

**Results on the KITTI DC online benchmark. (pttx, gNmQ).**

We report quantitative results using the same evaluation protocol as DesNet, the previous state-of-the-art on the KITTI DC benchmark. On average, ORCaS achieves a 3.7\% performance gain over DesNet (W1 in Part 2 response to pptx, W1b in Part 3 response to gNmQ).

**Ablation of the number of depth planes (pttx)**

We added an ablation study on the number of depth planes in ORCaS (pttx, Q4 in Part 1), showing that the performance improves as we add the number of depth planes from two to the proposed eight.

**Analysis on the ConteXt pooling size (ewac)**

As suggested by the reviewer ewac, we've conducted the ablation study on the ConteXt pooling size in the response (ewac, Q2 in Part 3): Overly small pooling size (2) lacks the receptive field to complete occluded regions, and the overly large pooling size (8) oversmooths the 3D voxels.

**Feature supervision vs. Depth supervision (ewac)**

We provided a comparison between feature supervision and depth supervision: Feature supervision yielded better results, which supported our choice. The result is shown in the response (ewac, W2a in Part 1).

**How to determine an appropriate frame interval to best assist depth completion learning? (pttx, gNmQ)**

We provide the results with longer temporal windows and their analysis. The result and the analysis are provided in the response (pttx, Q3 in Part 1, gNmQ, Q4a in Part 2). The result indicates that a larger temporal window improves the performance as long as the adjacent view has sufficient co-visible regions (and parallax) with a given view.

(2) Important questions from the reviewers.

**Theoretical Explanation of ORCaS (k3VB)**

We provided a response (k3VB, W1 in Part 1) that frames ORCaS as maximizing mutual information between the visible and occluded regions' representation by learning inductive bias.

** The improvements of outdoors vs. indoor datasets (gNmQ)**

While our improvement for outdoors (KITTI benchmark) is smaller than that of indoors, it is significant in that our gains over the current best method are more than the reported gains of previous state-of-the-art methods over their respective prior arts. This holds not only on the validation set (AugUndo, Tab. 6 in Supp. Mat) but also on the testing set (DesNet, W1b in pttx).

**Why is the alternating fashion for learning the occluded region needed? (gNmQ)**

We provided further explanation on the alternating fashion training (gNmQ, M2 in Part 1).

**How does the learning of the occluded region help the generalization on the current view? (pttx)**

We provided a further explanation (pttx, Q2 in Part 1) and mentioned the result in Tab. 4 in the Supp. Mat.

**How do we obtain camera poses for training? (pttx)**

Camera poses were predicted by a network. We explained in the response (pttx, Q1, Part 1).

(3) Typos, related works, and clarifications

We provided further explanations (especially for the typo in gNmQ, M2 in Part 1, and ewac, W3 in Part 2), and the items below will be revised accordingly.

- Missing related work on occluded scene reconstruction**

- The typo in the text description around Tab. 2 in the main paper**

All contents discussed during the rebuttal were incorporated into the revision of the manuscript, and the revisions are marked with blue text.

Paper 4651 Authors

---

### Meta-Review · Area_Chair_DhmW · 2026-01-11

**Summary:**

The paper proposes ORCaS, an unsupervised depth completion approach that learns to predict dense depth from RGB and sparse depth by introducing an auxiliary training objective: completing representations of regions that are occluded / unobserved in the input view using a nearby co-visible view. The method broadcasts 2D features into a 3D volume, warps features using relative pose, fills empty voxels with a ConteXt extrapolation module, and enforces feature consistency via an ORCaS loss. Across VOID1500 and NYUv2, the paper reports strong gains vs prior unsupervised methods, with additional cross-dataset generalization improvements and robustness under sparse inputs. During rebuttal, the authors also provided a more complete outdoor evaluation on KITTI DC online benchmark and requested ablations/clarifications. All reviewers explicitly remained positive after rebuttal.

**Reviewer Concerns:**

**Concerns addressed by rebuttal**
- Outdoor / KITTI evaluation strength and fairness of comparisons (pttx, gNmQ): Authors added results on KITTI DC online test benchmark following DesNet’s protocol; also clarified why initial appendix comparisons followed AugUndo’s reporting. Both reviewers stated concerns were addressed and kept accept scores.

- Ablations requested (pttx, ewac): Number of depth planes (MPI) ablation provided (D=2/4/8) showing monotonic improvements.
ConteXt pooling size ablation provided (2/4/8) and explanation of under/over-smoothing regimes. Feature supervision vs depth supervision experiment added; depth supervision was shown detrimental vs feature supervision.
- Training procedure clarity (gNmQ): The “alternating fashion” optimization was explained concretely (per-iteration alternation; which modules are frozen/updated).
- Theoretical intuition (k3VB): Authors provided a conceptual framing
- Compute / capacity concerns (ewac): Authors provided #params / GFLOPs / GPU memory / FPS comparisons and argued ORCaS remains real-time; also argued gains are not solely due to capacity.

**Concerns still only partially resolved:**
- Broader scalability beyond provided datasets: While KITTI online benchmark results help, reviewers originally asked for broader outdoor/high-variance testing. The rebuttal strengthens the claim but does not fully explore multiple large-scale outdoor datasets or diverse settings beyond KITTI.
- Qualitative emphasis on occluded regions (gNmQ): Authors pointed to Fig. 4 and supplementary evaluation on unobserved views, which helps, but a dedicated qualitative/metric breakdown explicitly targeting “occlusion-heavy” regions could further strengthen the narrative.
- Related work coverage up to 2024–2025 (k3VB, ewac): Authors promised to add more recent works; this is a straightforward camera-ready fix.

Overall: the remaining items are camera-ready polish rather than acceptance blockers.

**Reviewer Scores:**

- Reviewer pttx: 6, confidence 4 (reviewer explicitly: “concerns are addressed… keep my rating of acceptance.”)
- Reviewer gNmQ: 6, confidence 3  (explicitly: “Most concerns are addressed… keep my score to accept.”)
- Reviewer k3VB: 6, confidence 5 (explicitly voted for acceptance after rebuttal)
- Reviewer ewac: 6, confidence 4

---

### Decision · Program_Chairs · 2026-01-26

Accept (Poster)